# Assembling the Tat protein translocase

Felicity Alcock[1†], Phillip J Stansfeld[1*†], Hajra Basit[2‡], Johann Habersetzer[3], Matthew AB Baker[2§], Tracy Palmer[3], Mark I Wallace[2‡], Ben C Berks[1*]

[1]Department of Biochemistry, University of Oxford, Oxford, United Kingdom; [2]Department of Chemistry, University of Oxford, Oxford, United Kingdom; [3]Division of Molecular Microbiology, College of Life Sciences, University of Dundee, Dundee, United Kingdom

**Abstract** The twin-arginine protein translocation system (Tat) transports folded proteins across the bacterial cytoplasmic membrane and the thylakoid membranes of plant chloroplasts. The Tat transporter is assembled from multiple copies of the membrane proteins TatA, TatB, and TatC. We combine sequence co-evolution analysis, molecular simulations, and experimentation to define the interactions between the Tat proteins of *Escherichia coli* at molecular-level resolution. In the TatBC receptor complex the transmembrane helix of each TatB molecule is sandwiched between two TatC molecules, with one of the inter-subunit interfaces incorporating a functionally important cluster of interacting polar residues. Unexpectedly, we find that TatA also associates with TatC at the polar cluster site. Our data provide a structural model for assembly of the active Tat translocase in which substrate binding triggers replacement of TatB by TatA at the polar cluster site. Our work demonstrates the power of co-evolution analysis to predict protein interfaces in multi-subunit complexes.

*For correspondence: phillip. stansfeld@bioch.ox.ac.uk (PJS); ben.berks@bioch.ox.ac.uk (BCB)

[†]These authors contributed equally to this work

Present address: [‡]Department of Chemistry, Kings College London, London, United Kingdom; [§] EMBL Australia Node for Single Molecule Science, University of New South Wales, New South Wales, Australia

Competing interests: The authors declare that no competing interests exist.

## Introduction

Protein export across the cell membrane of prokaryotes occurs through two parallel pathways. Protein transport by the Sec apparatus involves a threading mechanism and requires the substrate protein to be maintained in an unstructured state (*Park and Rapoport, 2012*). By contrast, the Tat (twin-arginine translocation) system transports substrate proteins that have already achieved a folded conformation (*Berks, 2015*; *Cline, 2015*). In prokaryotes the requirement for a functional Tat pathway varies with the organism and their growth environment (*Palmer and Berks, 2012*). However, even under permissive growth conditions, loss of the Tat pathway results in serious pleiotropic effects on major cellular processes including energy metabolism, nutrient acquisition, virulence, and formation of the cell envelope (*Berks et al., 2003*; *De Buck et al., 2008*; *Palmer and Berks, 2012*). The Tat transport system has been evolutionarily conserved in plant chloroplasts where it mediates protein import across the thylakoid membrane and is essential for the formation of a functional photosynthetic apparatus (*Celedon and Cline, 2013*).

Tat transport depends on small integral membrane proteins from the TatA and TatC families. Minimal Tat systems found in some organisms contain a single type of TatA protein and one type of TatC molecule. These Tat systems are assumed to be ancestral to the more common arrangement in which a second, functionally distinct member of the TatA family, called TatB, is also present. The best-studied Tat systems, found in *Escherichia coli* and spinach chloroplasts, are examples of Tat systems containing both TatA and TatB proteins. Many organisms also possess further TatA paralogs. For example, *E. coli* has a third TatA family member called TatE which is functionally equivalent to TatA but is present at much lower concentration in the cell and not essential for Tat transport (*Jack et al., 2001*; *Sargent et al., 1998*).

Proteins are targeted to the Tat system by N-terminal signal peptides bearing the eponymous pair of arginine residues (*Berks, 1996*; *Chaddock et al., 1995*; *Stanley et al., 2000*). In *E. coli* and chloroplasts the signal peptide is recognized at the membrane by a receptor complex containing multiple copies of both TatB and TatC (*Cline and Mori, 2001*; *Tarry et al., 2009*). Substrate binding to the TatBC complex leads to the recruitment and oligomerization of TatA protomers from a pool in the membrane to form the active translocation site (*Alcock et al., 2013*; *Dabney-Smith et al., 2006*; *Rose et al., 2013*).

Atomic resolution structures have recently been determined for representative TatA, TatB, and TatC proteins (*Hu et al., 2010*; *Ramasamy et al., 2013*; *Rodriguez et al., 2013*; *Rollauer et al., 2012*; *Zhang et al., 2014a, 2014b*) (*Figure 1A*). Members of the TatA family conserve a core of two helical elements comprising a hydrophobic transmembrane helix (TMH) followed immediately by an

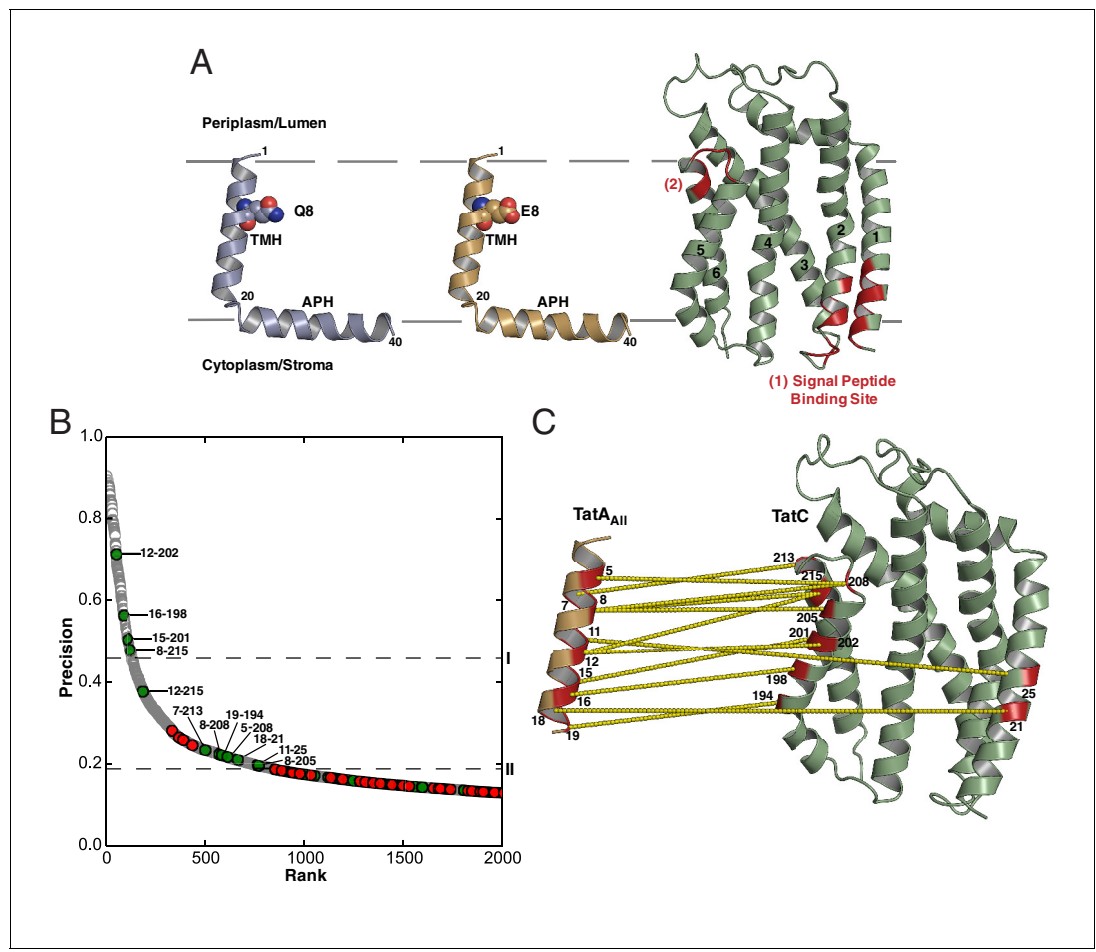

**Figure 1.** Sequence-coevolution analysis of interactions between TatA family proteins and TatC. (**A**) Structures of the *E. coli* Tat components. The transmembrane (TMH) and amphipathic (APH) helices of TatA and TatB are indicated. Areas of highest surface sequence conservation on TatC are indicated in red and include the binding site for the signal peptide twin-arginine motif. The natively unstructured tails of TatA and TatB are not depicted. (**B**) Predicted co-evolutionary residue contacts for the TatA_All–TatC dataset using the program PSICOV. Filled circles are predicted inter-subunit co-evolutionary contacts that are either (green) less than 15 Å apart along the membrane normal as expected of authentic direct contacts or (red) at greater than this value and therefore unlikely to correspond to direct interaction pairs. Unfilled gray circles are predicted intra-subunit contacts. Dashed line I marks the evolutionary coupling precision score (0.46) at 7SD above the mean for the whole dataset. Dashed line II marks the evolutionary coupling precision score (0.19) that is 6SD above the mean for the inter-subunit contact dataset. (**C**) A structural representation of the predicted TatA_All-TatC contacts above threshold level II. See also *Table 1*.

The following figure supplement is available for figure 1:

**Figure supplement 1.** Components of the Tat translocase within a lipid bilayer.

amphipathic helix (APH). TatC is composed of six transmembrane helices (TM1-TM6) shaped like a cupped hand.

Establishing how the multiple Tat components are arranged within the translocation complex is prerequisite for elucidating the mechanism of Tat transport. However, determining the structure of the Tat complexes by standard structural methods has proved to be exceedingly challenging due to the difficulties in producing suitable samples. Alternative structural approaches are therefore necessary.

Extensive efforts have been made to identify inter-subunit contacts within the Tat apparatus of both *E. coli* and chloroplasts using site-specific crosslinking. The most readily interpretable of these contacts suggest that the TMHs of TatA and TatB molecules interact with TatC toward the C-terminal end of the molecule. However, these contacts are by nature of low resolution and provide very limited information on the molecular structure of the interacting complexes. This situation is exacerbated by the possibility that a single TatC molecule is using multiple sites to simultaneously interact with different copies of the other Tat components. Two patches of highly conserved residues on the surface of TatC have previously been put forward as the sites of interaction with partner proteins (*Figure 1A*) (*Ramasamy et al., 2013*; *Rollauer et al., 2012*). The patch at the cytoplasmic side of the membrane has been identified as the binding site for the twin-arginine motif of the substrate signal peptide (*Holzapfel et al., 2007*; *Ma and Cline, 2013*; *Rollauer et al., 2012*) leaving the other patch at the periplasmic end of TatC TM5/TM6 as a plausible contact site for other components of the Tat apparatus. Again, this prediction provides no structural detail of the way the proteins interact. Finally, it has been speculated that a non-physiological helix-helix packing interaction at TM5 seen in crystals of *Aquifex aeolicus* TatC might mimic an interaction between TatC and the TMH of a TatA family member, with opinion divided as to whether this potential binding site would be occupied by TatA or TatB (*Aldridge et al., 2014*; *Cline, 2015*; *Rollauer et al., 2012*). This structural prediction remains to be definitively tested and does not provide an intrinsic molecular definition of the key components of the packing interface.

In an attempt to identify inter-protein contacts within the Tat apparatus with high precision, we turned to the emerging bioinformatics technique of sequence co-evolution analysis. The co-evolution approach relies on the principle that substitution of an amino acid at a tight packing interface will result in selection of compensatory changes in nearby amino acid side chains that re-optimize the interface. Thus, if two amino acids are in contact in the three-dimensional structure of a protein, sequence changes at one position will tend to be coupled with sequence changes at the other position. If these directly coupled amino acid changes can be identified from multiple sequence alignments, then it is possible to predict the packing interactions within a protein of unknown structure. Although most attention has been focused on applying co-evolution analysis in the de novo prediction of protein folds, recent studies show that it is also possible to identify inter-protein contacts using this method (*Dago et al., 2012*; *Hopf et al., 2014*; *Ovchinnikov et al., 2014*; *Wang and Barth, 2015*).

We have used sequence co-evolution methods to provide a picture of protein-protein interactions within the Tat system that is independent of previously employed methodologies and is of sufficiently high resolution to allow explicit molecular modeling of the multi-subunit TatBC receptor complex. Key elements of the structural prediction have been experimentally verified including the discovery of a functionally crucial intramembrane cluster of polar amino acids. Our results allow us to address how the two structurally related proteins TatA and TatB can have different patterns of interaction with TatC. They also suggest how substrate binding to the TatBC complex triggers TatA oligomerization and formation of the Tat translocation site. Our work highlights the potential of sequence co-evolution analysis to provide accurate molecular-level information on the contact interfaces within complex multi-protein complexes.

## Results

### Evolutionary contacts between TatA family proteins and TatC

We applied sequence co-evolution analysis to characterize the interactions that exist between TatA family proteins and TatC. To maximize the sensitivity of the analysis, we initially combined all available TatA family sequences into a single data set ('TatA$_{All}$') in order to fully utilize the available

sequence diversity. The resulting analysis included 6002 non-identical TatA$_{All}$-TatC sequence pairs. The evolutionary coupling between each pair of positions has a precision score between 0 and 1 which indicates the probability that a genuine evolutionary coupling has been detected. Thus, evolutionary couplings with precision scores above 0.5 are more likely to represent a real coupling between the residue pair than they are to be a false positive.

Ten out of the 12 most highly co-evolving sequence pairs, including three with precision scores above 0.5, are directed from one face of the TatA TMH toward surface-exposed residues in TM5 and TM6 of TatC (*Figure 1B,C* and *Table 1*). This suggests that TatA family proteins bind to the C-terminal end of TatC. The evolutionarily coupled residues lie in the same order along the faces of the two proteins as expected of an authentic contact interface between TM helices. It is also notable that the unusually short TatA TMH is well-matched in length to the section of the kinked TatC TM5 with which it primarily interacts.

The majority of the contacts at the inferred interface are between amino acids with hydrophobic side-chains. However, residue eight toward the periplasmic end of the TatA TMH normally has a polar side chain (Q8 in *E. coli* TatA, E8 in *E. coli* TatB, K8 in *E. coli* TatE) and this residue is involved in three predicted contacts with residues corresponding to M205, T208 and Q215 in *E. coli* TatC. Notably, the latter residues fall within the functionally unassigned patch of highly conserved surface residues at the C-terminal end of TatC (*Figure 1A*, region (2)).

## Exploring evolutionary contacts for different TatA paralogs

Our co-evolution analysis using all members of the TatA family strongly predicts an interaction between the TatA family TMH and TM5/6 of TatC. However, in those organisms that use both a TatA and a TatB protein, these two TatA paralogs cannot simultaneously occupy the same binding site on TatC. Additionally, the current model for the Tat transport cycle suggests that TatA and TatB have non-identical interactions with TatC (*Berks, 2015*; *Cline, 2015*). These considerations raise the possibility that our TatC contact site prediction arises from a subset of the TatA paralogs within the TatA$_{All}$ dataset and that other TatA paralogs do not bind at the predicted interaction site. We therefore repeated the co-evolution analysis using sequence subsets representing different members of the TatA protein family.

Assigning individual TatA protein sequences to specific subsets presents significant challenges because no specific sequence motifs allow unambiguous separation of TatB proteins from other TatA paralogs (*Berks, 2015*). It is also unclear what sequence criteria differentiate the TatA proteins of minimal Tat systems from the functionally differentiated TatA paralogs found in TatABC systems. In an attempt to address these issues, we sorted the proteins in the TatA$_{All}$ dataset by sequence similarity across the entire TMH and APH regions thereby maximizing the sequence information used for the comparison. A phylogenetic tree based on this analysis reveals three distinct and well-separated groupings (*Figure 2A*). The central grouping contains proteins from organisms containing only a single TatA family molecule (corresponding to organisms with a minimal Tat system) or multiple, closely related TatA family molecules. We designate this sequence subset the 'TatA$_{basal}$' dataset (1344 sequences) to reflect the position of these proteins at the root of the tree. The remaining sequences fall into two groupings that diverge in opposite directions from the TatA$_{basal}$ dataset. Because *E. coli* TatB falls in one of the additional groupings, and *E. coli* TatA and TatE proteins in the other, we designate these sequence groupings the TatB dataset (3883 sequences) and the TatA dataset (6010 sequences). Ninety-four percent of organisms with a TatB protein also have a TatA protein. However, 26% of the organisms possessing TatA proteins do not have a TatB protein, with this situation being more common for TatA proteins that have low divergence from TatA$_{basal}$. TatA proteins from organisms that did not also have a TatB protein were removed from the TatA dataset to ensure that only genuine TatABC systems were included in our analysis (4625 of the TatA sequences were retained).

Given the limited number of Tat systems for which biochemical data are available, it is not certain that our three TatA sequence subsets correspond exactly to different functional categories. Nevertheless, the TatA and TatB subsets are representative of the two distinct structural subclasses into which the well-characterized *E. coli* TatA and TatB proteins fall.

We produced sequence logos for each TatA family dataset to reveal sequence features that are strongly associated with the three different structural subsets (*Figure 2B*). The TatA family subsets differ in which of the strongly conserved features of the TatA$_{basal}$ proteins they retain. Residue eight

**Table 1.** Comparison of PSICOV predictions for Tat component contacts with those of other co-evolutionary methods. The programs PSICOV (*Jones et al., 2012*), CCMPRED (*Seemayer et al., 2014*), and FreeContact (mfDCA) (*Hopf et al., 2012*; *Kaján et al., 2014*) use methodologically distinct algorithms to calculate sequence co-evolution. Contacts predicted by all three of the co-evolutionary methods are colored green. Meta-PSICOV (*Jones et al., 2015*) aggregates information from the other three prediction methods as well as other sequence information in Stage 1, then weights by vicinity to other contacts in Stage 2. The table is divided to include the co-evolutionary contacts for TatA$_{All}$, TatA, TatB, TatA with TatA$_{basal}$, TatA with TatB$_{basal}$, and TatC-TatC contacts with separations greater than 20 Å. For the TatC-TatC contacts the three high-probabilty contacts identified by the three distinct methods suggest plausible contacts between the TM2-3 linker and residues at the C-terminal end of TM3 (green).

| | PSICOV | | | | FreeContact (mfDCA) | | | | CCMPRED | | | | Meta-PSICOV Stage 1 | | | | Meta-PSICOV Stage 2 | | | |
|---|---|---|---|---|---|---|---|---|---|---|---|---|---|---|---|---|---|---|---|---|
| | TMH | TatC | Score | Rank | TMH | TatC | Score | Rank | TMH | TatC | Score | Rank | TMH | TatC | Score | Rank | TMH | TatC | Score | Rank |
| TatA$_{All}$ | 12 | 202 | 0.71 | 51 | 12 | 202 | 1.46 | 108 | 12 | 202 | 0.30 | 28 | 12 | 202 | 0.75 | 22 | 12 | 202 | 0.46 | 112 |
| | 16 | 198 | 0.56 | 88 | 15 | 201 | 1.00 | 305 | 16 | 198 | 0.27 | 59 | 16 | 198 | 0.71 | 30 | 15 | 201 | 0.40 | 145 |
| | 15 | 201 | 0.5 | 109 | 16 | 198 | 0.95 | 335 | 15 | 201 | 0.23 | 117 | 15 | 201 | 0.64 | 49 | 16 | 198 | 0.34 | 191 |
| | 8 | 215 | 0.48 | 123 | 8 | 215 | 0.84 | 422 | 12 | 215 | 0.21 | 175 | 12 | 215 | 0.19 | 414 | 12 | 201 | 0.15 | 444 |
| | 12 | 215 | 0.38 | 189 | 12 | 215 | 0.75 | 549 | 18 | 21 | 0.19 | 313 | 15 | 198 | 0.14 | 609 | 12 | 198 | 0.08 | 771 |
| | 7 | 213 | 0.23 | 497 | 18 | 21 | 0.56 | 1010 | 11 | 212 | 0.18 | 323 | 18 | 21 | 0.14 | 611 | 8 | 215 | 0.05 | 1171 |
| | 8 | 208 | 0.22 | 578 | 8 | 208 | 0.53 | 1128 | 8 | 215 | 0.18 | 388 | 8 | 215 | 0.11 | 734 | 4 | 205 | 0.04 | 1332 |
| | 19 | 194 | 0.22 | 591 | 14 | 172 | 0.45 | 1577 | 15 | 198 | 0.18 | 400 | 8 | 208 | 0.09 | 860 | 8 | 205 | 0.04 | 1593 |
| | 5 | 208 | 0.22 | 619 | 8 | 205 | 0.44 | 1630 | 5 | 208 | 0.18 | 404 | 12 | 201 | 0.06 | 1072 | 5 | 208 | 0.03 | 1726 |
| | 18 | 21 | 0.21 | 670 | 5 | 126 | 0.43 | 1717 | 11 | 174 | 0.16 | 674 | 12 | 198 | 0.06 | 1083 | 15 | 198 | 0.03 | 1897 |
| | 11 | 25 | 0.2 | 768 | 15 | 198 | 0.42 | 1761 | 8 | 208 | 0.15 | 753 | 14 | 203 | 0.05 | 1175 | 8 | 208 | 0.03 | 1963 |
| | 8 | 205 | 0.19 | 772 | 17 | 85 | 0.42 | 1851 | 4 | 205 | 0.15 | 793 | 4 | 205 | 0.05 | 1176 | 11 | 202 | 0.03 | 2044 |
| | 8 | 166 | 0.19 | 855 | 5 | 124 | 0.39 | 2105 | 17 | 227 | 0.15 | 802 | 5 | 208 | 0.04 | 1367 | 8 | 198 | 0.02 | 2299 |
| | 12 | 198 | 0.18 | 911 | 5 | 132 | 0.38 | 2131 | 19 | 194 | 0.15 | 960 | 11 | 25 | 0.04 | 1424 | 15 | 202 | 0.02 | 2569 |
| | 17 | 227 | 0.17 | 1003 | 12 | 194 | 0.33 | 2870 | 8 | 214 | 0.15 | 979 | 11 | 174 | 0.04 | 1544 | 4 | 208 | 0.02 | 2846 |
| | 15 | 198 | 0.17 | 1150 | 12 | 75 | 0.32 | 2995 | 12 | 198 | 0.15 | 1036 | 15 | 219 | 0.03 | 1574 | 8 | 202 | 0.02 | 3287 |
| | 12 | 201 | 0.16 | 1183 | 12 | 198 | 0.32 | 3008 | 12 | 201 | 0.14 | 1158 | 10 | 25 | 0.03 | 1607 | 4 | 206 | 0.01 | 3437 |
| TatA | 12 | 202 | 0.71 | 18 | 12 | 202 | 1.81 | 16 | 12 | 202 | 0.21 | 10 | 12 | 202 | 0.72 | 12 | 12 | 202 | 0.46 | 91 |
| | 16 | 198 | 0.55 | 39 | 16 | 198 | 1.12 | 124 | 16 | 198 | 0.19 | 15 | 16 | 198 | 0.72 | 13 | 9 | 206 | 0.27 | 229 |
| | 14 | 216 | 0.38 | 122 | 12 | 215 | 1.06 | 155 | 9 | 206 | 0.14 | 49 | 9 | 206 | 0.40 | 82 | 16 | 198 | 0.27 | 230 |
| | 16 | 82 | 0.37 | 133 | 5 | 136 | 0.91 | 246 | 15 | 163 | 0.14 | 67 | 15 | 163 | 0.35 | 98 | 4 | 206 | 0.18 | 378 |
| | 9 | 206 | 0.34 | 162 | 5 | 210 | 0.86 | 287 | 3 | 205 | 0.14 | 72 | 15 | 202 | 0.29 | 137 | 9 | 202 | 0.11 | 630 |
| | 5 | 162 | 0.29 | 262 | 15 | 163 | 0.86 | 297 | 12 | 215 | 0.14 | 74 | 16 | 82 | 0.28 | 146 | 8 | 206 | 0.09 | 830 |
| | 5 | 136 | 0.29 | 270 | 8 | 212 | 0.77 | 408 | 13 | 197 | 0.14 | 79 | 15 | 201 | 0.24 | 184 | 5 | 210 | 0.09 | 835 |
| | 9 | 39 | 0.28 | 290 | 19 | 170 | 0.74 | 470 | 16 | 82 | 0.14 | 86 | 19 | 20 | 0.20 | 271 | 15 | 202 | 0.09 | 866 |
| | 12 | 215 | 0.26 | 339 | 15 | 202 | 0.72 | 530 | 2 | 219 | 0.13 | 121 | 12 | 215 | 0.18 | 317 | 18 | 232 | 0.09 | 880 |
| TatB | 18 | 21 | 0.83 | 7 | 18 | 21 | 2.19 | 6 | 18 | 21 | 0.23 | 9 | 12 | 198 | 0.72 | 12 | 12 | 198 | 0.61 | 41 |
| | 12 | 215 | 0.60 | 30 | 12 | 215 | 1.45 | 22 | 12 | 198 | 0.18 | 14 | 12 | 202 | 0.63 | 23 | 12 | 202 | 0.47 | 97 |
| | 12 | 202 | 0.56 | 36 | 12 | 198 | 1.39 | 24 | 18 | 24 | 0.17 | 16 | 18 | 21 | 0.61 | 29 | 18 | 21 | 0.41 | 125 |
| | 20 | 185 | 0.46 | 65 | 12 | 202 | 1.13 | 63 | 12 | 202 | 0.16 | 17 | 18 | 24 | 0.53 | 47 | 18 | 24 | 0.32 | 194 |
| | 12 | 198 | 0.39 | 110 | 19 | 13 | 1.04 | 89 | 12 | 215 | 0.16 | 21 | 12 | 215 | 0.49 | 55 | 5 | 212 | 0.20 | 350 |
| | 5 | 213 | 0.36 | 127 | 5 | 212 | 0.97 | 117 | 17 | 227 | 0.14 | 71 | 19 | 13 | 0.39 | 79 | 12 | 215 | 0.20 | 355 |
| | 7 | 204 | 0.35 | 141 | 5 | 126 | 0.87 | 164 | 14 | 167 | 0.14 | 72 | 14 | 24 | 0.25 | 175 | 4 | 206 | 0.16 | 462 |
| | 5 | 212 | 0.28 | 281 | 18 | 24 | 0.81 | 221 | 5 | 213 | 0.13 | 88 | 19 | 198 | 0.22 | 225 | 5 | 208 | 0.14 | 529 |

*Table 1 continued on next page*

Table 1 continued

| | PSICOV | | | | FreeContact (mfDCA) | | | | CCMPRED | | | | Meta-PSICOV Stage 1 | | | | Meta-PSICOV Stage 2 | | | |
|---|---|---|---|---|---|---|---|---|---|---|---|---|---|---|---|---|---|---|---|---|
| | TMH | TatC | Score | Rank | TMH | TatC | Score | Rank | TMH | TatC | Score | Rank | TMH | TatC | Score | Rank | TMH | TatC | Score | Rank |
| TatA with TatA_basal | 12 | 202 | 0.75 | 28 | 12 | 202 | 1.85 | 26 | 16 | 198 | 0.31 | 10 | 16 | 198 | 0.80 | 8 | 12 | 202 | 0.47 | 94 |
| | 16 | 198 | 0.70 | 35 | 16 | 198 | 1.26 | 62 | 12 | 202 | 0.30 | 11 | 12 | 202 | 0.77 | 14 | 16 | 198 | 0.43 | 111 |
| | 15 | 201 | 0.60 | 58 | 8 | 215 | 1.01 | 130 | 15 | 201 | 0.22 | 40 | 15 | 201 | 0.68 | 36 | 15 | 201 | 0.39 | 136 |
| | 8 | 215 | 0.58 | 64 | 15 | 201 | 1.01 | 131 | 11 | 212 | 0.17 | 126 | 8 | 215 | 0.18 | 309 | 12 | 201 | 0.08 | 644 |
| | 19 | 194 | 0.38 | 141 | 12 | 215 | 0.69 | 378 | 8 | 215 | 0.16 | 143 | 14 | 203 | 0.14 | 419 | 8 | 215 | 0.06 | 832 |
| | 8 | 205 | 0.27 | 309 | 5 | 136 | 0.67 | 426 | 12 | 215 | 0.16 | 159 | 8 | 208 | 0.09 | 605 | 4 | 205 | 0.06 | 852 |
| TatB with TatA_basal | 18 | 21 | 0.48 | 72 | 18 | 21 | 1.42 | 44 | 18 | 21 | 0.23 | 19 | 12 | 198 | 0.65 | 36 | 12 | 198 | 0.48 | 94 |
| | 12 | 198 | 0.46 | 79 | 12 | 198 | 0.99 | 173 | 12 | 198 | 0.19 | 37 | 12 | 202 | 0.56 | 55 | 4 | 206 | 0.43 | 119 |
| | 12 | 202 | 0.38 | 123 | 12 | 202 | 0.84 | 281 | 12 | 202 | 0.19 | 39 | 18 | 21 | 0.56 | 56 | 18 | 21 | 0.28 | 214 |
| | 20 | 185 | 0.37 | 134 | 12 | 215 | 0.76 | 383 | 18 | 24 | 0.18 | 44 | 15 | 201 | 0.34 | 122 | 12 | 202 | 0.28 | 218 |
| | 12 | 215 | 0.33 | 180 | 15 | 201 | 0.73 | 427 | 14 | 167 | 0.17 | 69 | 4 | 206 | 0.28 | 168 | 4 | 207 | 0.28 | 225 |
| | 5 | 213 | 0.30 | 252 | 5 | 212 | 0.68 | 514 | 5 | 213 | 0.17 | 88 | 18 | 24 | 0.27 | 175 | 4 | 205 | 0.16 | 415 |
| TatC-TatC | 64 | 134 | 0.79 | 33 | 27 | 199 | 2.04 | 39 | 64 | 134 | 0.28 | 51 | 64 | 134 | 0.6 | 62 | 64 | 134 | 0.64 | 60 |
| | 71 | 140 | 0.77 | 41 | 65 | 137 | 1.72 | 62 | 71 | 140 | 0.26 | 75 | 71 | 140 | 0.54 | 84 | 27 | 199 | 0.47 | 108 |
| | 65 | 137 | 0.64 | 69 | 71 | 140 | 1.63 | 72 | 65 | 137 | 0.21 | 191 | 27 | 199 | 0.48 | 103 | 65 | 137 | 0.3 | 227 |
| | 28 | 203 | 0.51 | 10864 | 64 | 134 | 1.51 | 93 | 113 | 228 | 0.19 | 319 | 65 | 137 | 0.39 | 150 | 64 | 137 | 0.24 | 283 |
| | 72 | 217 | 0.42 | 155 | 64 | 157 | 1.05 | 284 | 67 | 140 | 0.18 | 345 | 28 | 203 | 0.37 | 162 | 33 | 167 | 0.23 | 289 |

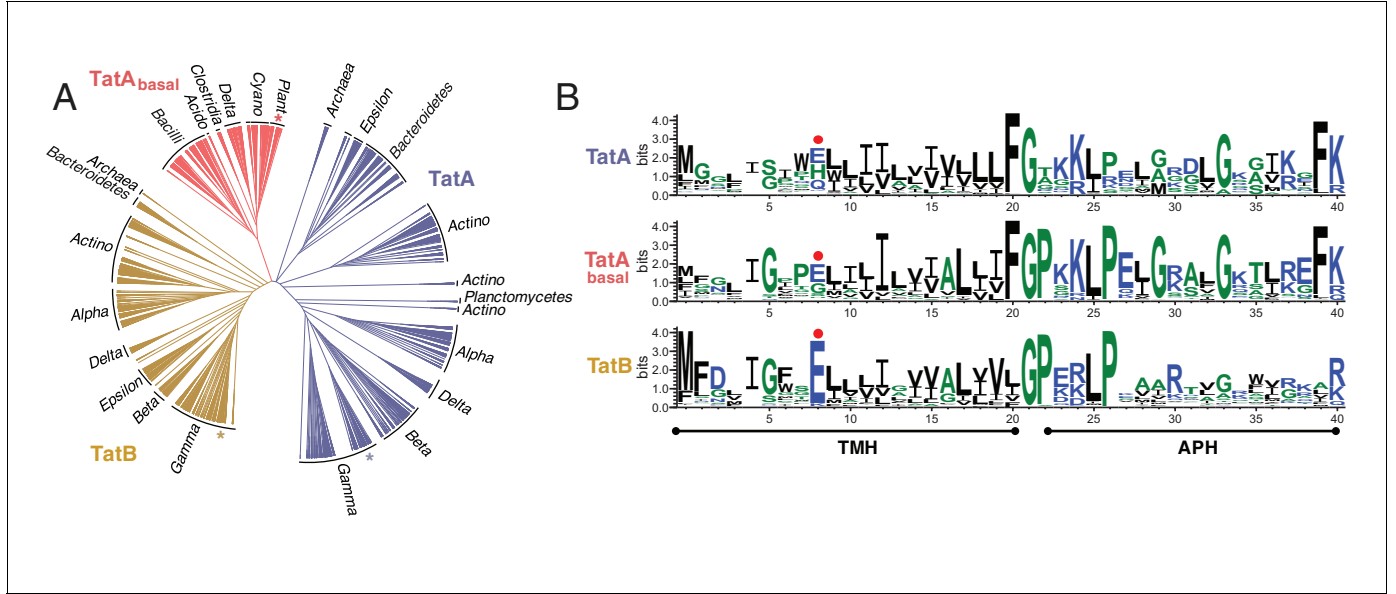

**Figure 2.** Phylogeny and sequence conservation of the TatA family. (**A**) Phylogenetic tree of TatA family members based on a sequence comparison to the end of the APH. The phylogenetic origins of the sequences are given around the edge of the tree: '*Alpha*', '*Beta*', '*Gamma*', '*Delta*', and '*Epsilon*' refer to the five classes of the phylum Proteobacteria; '*Bacilli*' and '*Clostridia*' are classes within the phylum Firmicutes; '*Actino*' refers to the phylum *Actinobacteria*; '*Acido*' refers to the phylum *Acidobacteria*; '*Cyano*' refers to the phylum *Cyanobacteria*; 'Plants' refers to plant chloroplast proteins. Asterisks mark the sequences of *E. coli* TatA/TatE (blue), *E. coli* TatB (orange), and the pea thylakoid TatA and TatB proteins Tha4 and Hcf106 (red). (**B**) Sequence logos for the three Tat subsets identified in (**A**). The logos correspond to the sequence region analyzed in (**A**) and use the sequence numbering of *E. coli* TatA and TatB. The secondary structure elements of the Tat proteins are shown under the logos and the position of the TMH polar residue is indicated with a red dot. The figure was generated using WebLogo 3.4 (**Crooks et al., 2004**).

is most commonly a glutamic acid in all TatA family proteins. Whilst this amino acid is almost always present in TatB proteins, TatA proteins often substitute the alternative polar amino acids histidine or glutamine, and TatA$_{basal}$ proteins use glycine or serine at significant frequency. The conserved glycine residue that separates the TMH and APH (G21 in *E. coli* TatA and TatB) is found in distinctive sequence contexts in the three TatA family subsets. In TatA proteins the motif is normally F-G-X, in TatB proteins X-G-P, while TatA$_{basal}$ proteins combine both motifs in the form of a F-G-P sequence. Within the APH TatB and TatA$_{basal}$ proteins much more strongly conserve a L-P dipeptide at residues 25 and 26 than TatA proteins do, whilst the TatA proteins and TatA$_{basal}$ proteins conserve a glycine at position 33 more strongly than TatB proteins. Finally, the almost invariant and functionally essential phenylalanine residue at the end of the APH of TatA and TatA$_{basal}$ (F39 in *E. coli* TatA) (*Alcock et al., 2013*; *Hicks et al., 2003*, *2005*) is normally absent from TatB proteins.

We performed co-evolution analyses between each TatA family sequence subset and the TatC sequences from the same organisms. Although the TatA and TatB comparisons were less sensitive than the earlier analysis using the TatA$_{All}$ dataset, they both retain some of the contact pairs at the TMH-TatC TM5/TM6 interface with precision scores >0.5 (*Figure 3—figure supplement 1*). Thus, the co-evolution analysis predicts that both classes of TatA paralog bind to the same site at the C-terminal end of TatC. Co-evolution analysis using the TatA$_{basal}$ dataset did not identify statistically significant contacts with TatC. However, when the TatA$_{basal}$ dataset was combined with the TatA dataset, more contacts were recovered at the TM5/TM6 site than were detected using the TatA dataset alone (*Figure 3—figure supplement 1*). This observation implies that TatA$_{basal}$ also binds at the TatC TM5/TM6 site.

## Molecular modeling of the primary TatAC contact site

We used the contact predictions from the co-evolution analyses to build molecular models for the interaction of *E. coli* TatC with the TMHs of *E. coli* TatA and *E. coli* TatB. A homology model of *E. coli* TatC was generated from the crystal structure of *A. aeolicus* TatC. The initial positions of the TatA and TatB TMHs relative to TatC were based on the position of the inverted TatC TM5 found at the packing interface in *A. aeolicus* TatC crystals (*Ramasamy et al., 2013*; *Rollauer et al., 2012*) because the position of this helix resembles the location of the TatA family TMH predicted by co-evolution analysis. The resulting models are shown in *Figure 3*. In an alternative approach, the TatA and TatB TMHs were docked to TatC using the TatA$_{All}$ evolutionary couplings as unambiguous restraints in the program Haddock (*Dominguez et al., 2003*) and the output models then ranked by consistency with the co-evolution analysis. The docking-derived models agreed well with the crystal packing-based models with RMSDs of less than 0.5 Å. The docked positions of the two TMHs were also very similar with a Cα-RMSD of 0.2 Å between TatA and TatB.

Atomistic molecular dynamics (MD) simulations in a membrane bilayer environment were used to assess and optimize the modeled TatAC and TatBC interfaces (*Video 1*). The TatBC interface was simulated with E8 in either the protonated or deprotonated state. For each model we assessed the Cα-Cα distances between the evolutionary coupled residue pairs over the course of the simulations (*Figure 4A*). After minor equilibration of the interaction interface, the simulations settled to a stable state in which the two proteins remained tightly packed and where the secondary structure in the starting models was preserved (*Figure 4A* and *Figure 4—figure supplement 1*). The stability of the models provides computational support for the interaction site suggested by evolutionary methods. It also confirms that the contact site on *E. coli* TatC could plausibly bind either TatA or TatB.

Detailed examination of the simulations shows that the polar residue found at position eight in TatA and TatB participates in hydrogen-bonding interactions with some of its strongly co-evolving partner residues in TatC (*Figure 4C* and *Video 2*). In the case of TatB, the carboxylate side chain of deprotonated E8 acts as a hydrogen bond acceptor from both TatC T208 and Q215. These residue interactions are maintained when TatB E8 is protonated but now TatC T208 is the hydrogen bond acceptor from TatB E8. TatC Q215 also acts as a hydrogen bond donor to TatA Q8. However, no hydrogen bonds are formed between TatA Q8 and TatC T208 in any of the simulations. Additional stabilization of the position of the TatA/TatB polar residue side chain arises from aliphatic interactions with the strongly co-evolving TatC residue M205. Sequence analysis shows that other amino acid pairs commonly found at position eight in TatA family proteins and position 215 in TatC have the potential to hydrogen bond with each other (e.g. H8-Q215, K8-Q215, S8-H215). This suggests

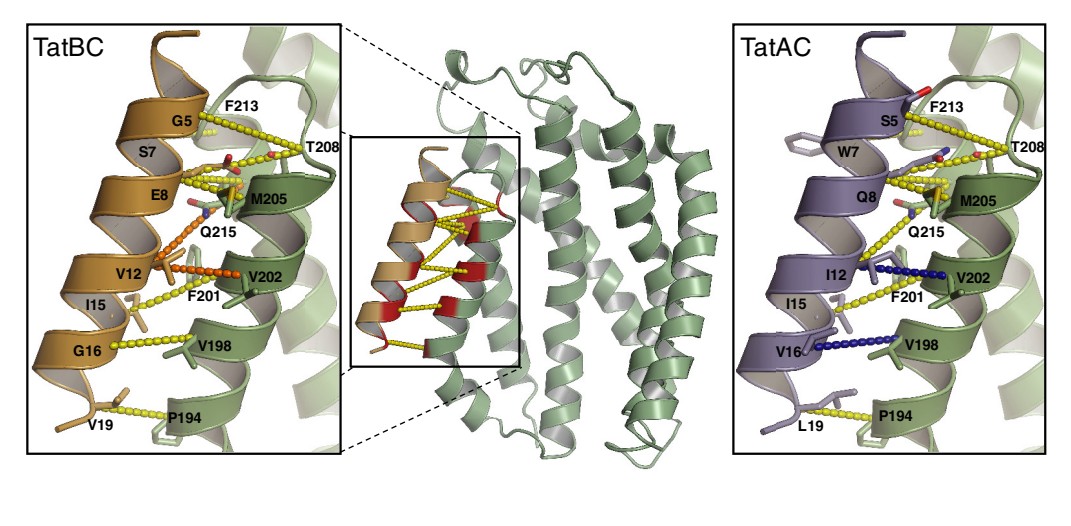

**Figure 3.** Modeling the interaction of TatA and TatB with the C-terminal end of TatC. The interaction between the TatA/B TMH and TatC TM5/TM6 modeled for *E. coli* TatBC and TatAC pairs. The models are based on *A. aeolicus* TatC crystal packing contacts. Evolutionary couplings for the TatA$_{All}$ dataset are shown (dotted lines). Couplings retained in just the TatB or TatA datasets with precisions greater than 0.5 (*Figure 3—figure supplement 1A*) are colored orange or blue, respectively. See also (*Table 1*).

The following figure supplement is available for figure 3:

**Figure supplement 1.** Co-evolution analysis of inter-subunit contacts in the Tat system.

that a polar interaction between the residues at these positions is a general feature of the Tat system.

## The polar cluster is essential for Tat transport

The prediction that TatA family proteins interact with TatC through an intramembrane cluster of polar amino acid residues is noteworthy because polar contacts between membrane-spanning helices are rare, and normally of functional significance (*Popot and Engelman, 2000*). This led us to explore whether perturbations of the predicted polar cluster could be used to test the co-evolution-derived model for Tat component interactions.

We used the *E. coli* Tat system as our experimental model with the Tat proteins expressed at their native levels. Strains are named for the Tat proteins they produce. Thus, the wild-type strain is called 'ABCE' and contains all of TatA, TatB, TatC, together with the TatA paralog TatE. For technical convenience, TatB and TatC variants were expressed from the low copy number plasmid p101C*TatBC (*Alcock et al., 2013*) in a Δ*tatBC* background (strain MΔBC) to give a strain designated 'AE pBC'. Plasmid p101C*TatBC directs expression of TatB and TatC at native levels and fully restores Tat transport activity to a Δ*tatBC* strain (*Figure 5—figure supplement 1A*).

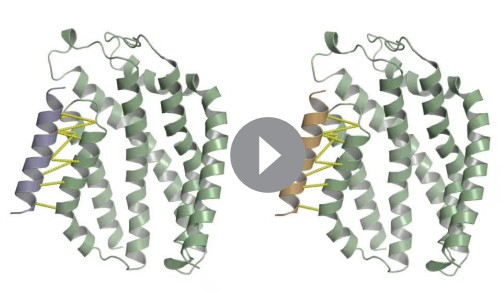

**Video 1.** Representative atomistic molecular dynamics simulations of *E. coli* TatA-TatC and TatB-TatC heterodimers in a phospholipid bilayer. The interactions of the wild-type TatA TMH (blue; left panel panel) or TatB TMH (deprotonated E8) (orange; right hand panel) with TatC (green) were assessed in a 100 ns molecular simulation. Yellow dashed lines connect the predicted co-evolving residue pairs shown in *Figure 3*.

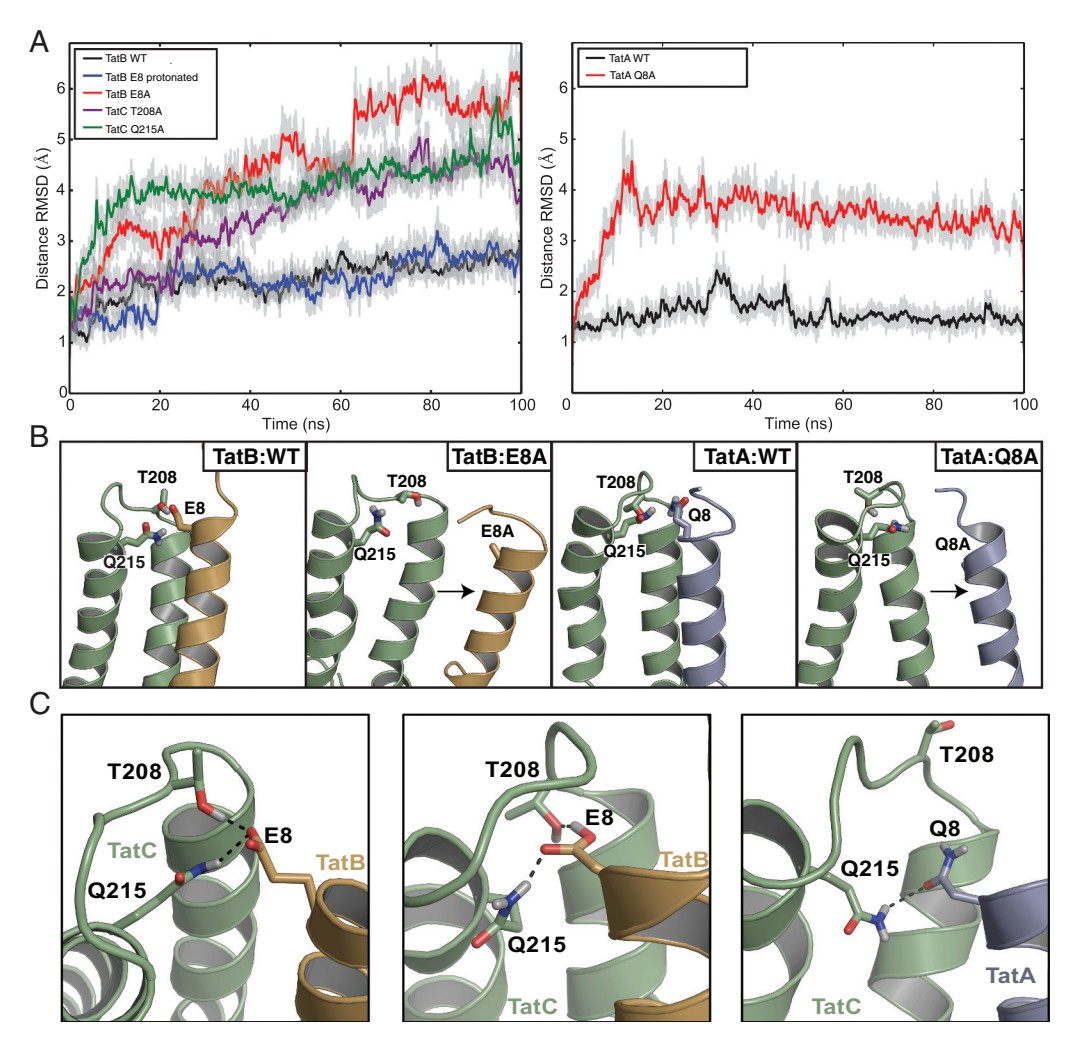

**Figure 4.** Molecular simulations of the interactions of TatA and TatB with the C-terminal end of TatC. (A) Root-mean square deviation (RMSD) of the distances between predicted contact pairs during atomistic MD simulations of the indicated TatBC (*left*) and TatAC (*right*) models in a membrane environment taken from three simulations. Both raw data (light gray) and data averaged over a rolling window of 0.35 ns (bold) are shown. Except where indicated, TatB E8 was deprotonated in the simulations. (B) Alanine substitution of the TMH polar residue disrupts the interaction between TatC and the TMHs of TatB (orange) or TatA (blue). The output structures from 100 ns MD simulations are shown with the helix displacements seen in the variants (right hand panel in each pair) relative to the wild-type proteins (left hand panel in each pair) denoted by arrows. (C) Snapshots of the MD simulations of the TatBC and TatAC models showing hydrogen bonding interactions between residues in the inter-subunit polar cluster. Simulations were run with TatB E8 either deprotonated (left panel) or protonated (center panel). See also *Videos 1–2*.

The following figure supplement is available for figure 4:

**Figure supplement 1.** Structural stability plots for the modeled Tat protein complexes from molecular simulations.

To test whether the predicted polar cluster is important for Tat function we analyzed alanine substitutions of the cluster-forming residues, namely TatA Q8, TatB E8, or TatC M205, T208, and Q215. When examining the functional effects of the TatA Q8A substitution we used a strain lacking both TatA and its paralog TatE.

We used two methods to assess the transport ability of the variants. Firstly, we overproduced the Tat substrate CueO and determined how much of this protein reached the periplasm. Under these conditions of substrate saturation the export of CueO is proportional to the transport capacity of the Tat pathway. Secondly, we characterized the ability of the variant Tat proteins to correct the

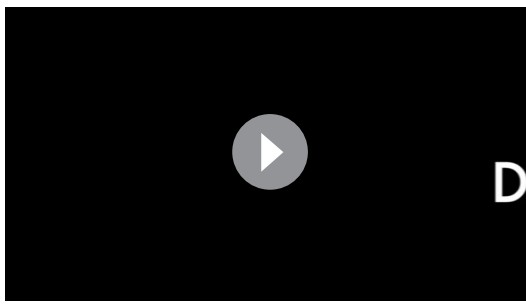

**Video 2.** Polar cluster interactions between TatC residues T208 and Q215 (green) and either deprotonated TatB residue E8 (orange), or protonated TatB residue E8 (orange), or TatA residue Q8 (blue) during 100 ns of molecular simulation. Black dashed lines indicate hydrogen bonds.

cell-chaining phenotype associated with a defective Tat pathway. This cell-chaining behavior arises from the mislocalization of Tat-targeted periplasmic amidases involved in splitting the septal murein after cell division (*Bernhardt and de Boer, 2003*; *Ize et al., 2003*). Cell chaining is only observed in cells with an almost completely non-functional Tat pathway.

Neither the TatA Q8A nor TatB E8A variants supported detectable CueO export (*Figure 5A*; strains $A^{Q8A}BC$ and AE $pB^{E8A}C$). Cells of the TatB variant were also fully chained indicating a complete absence of Tat transport (*Figure 5B* and *Figure 5—figure supplement 1*; strain AE $pB^{E8A}C$) whilst cultures of the TatA variant contained both single cells and short chains suggesting retention of a very low level of Tat function (*Figure 5B*; strain $A^{Q8A}BC$). Although the

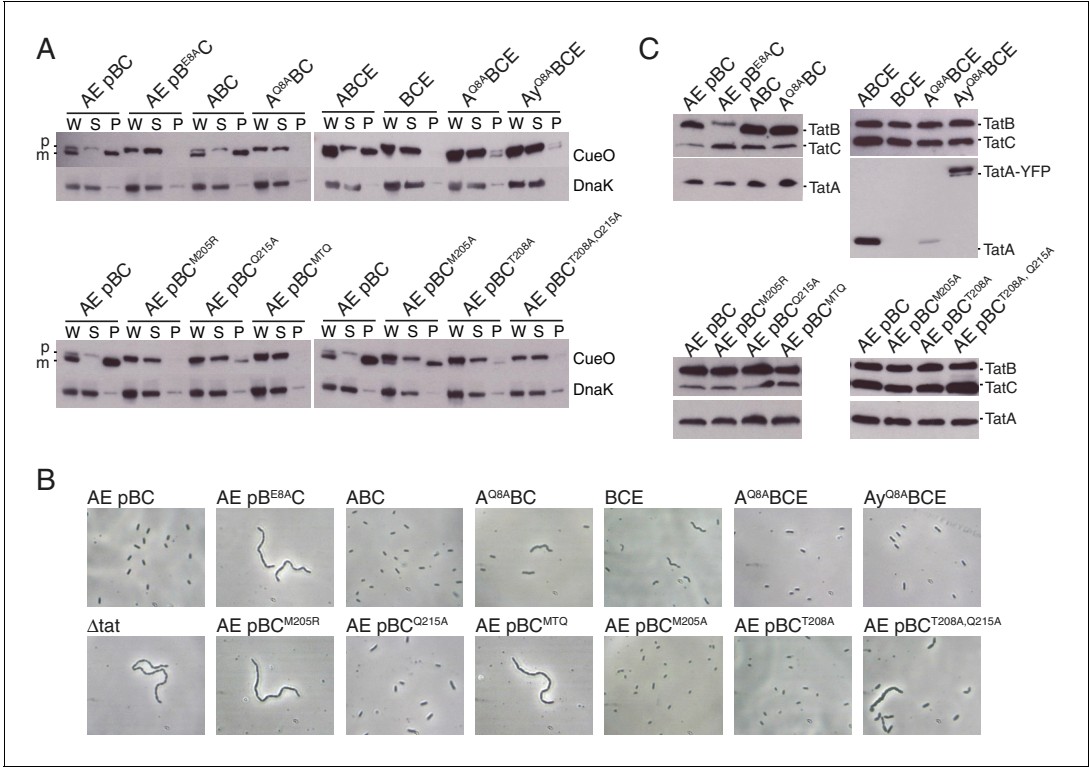

**Figure 5.** Polar cluster substitutions impair Tat transport. Strains contained the indicated amino acid substitutions in chromosomally encoded TatA or plasmid-encoded TatB or TatC. $TatC^{MTQ}$ is a combination of the three substitutions M205A, T208A, and Q215A. TatAy is TatA with a C-terminal fusion to yellow fluorescent protein. (**A**) Transport activity of strains overproducing the Tat substrate CueO. Whole cell (W), spheroplast (S) and periplasm (P) fractions were subject to immunoblotting with antibodies against CueO or the cytoplasmic marker protein DnaK. m is the transported form of CueO from which the signal peptide has been removed and p the precursor protein. (**B**) Phase contrast images of the strains. 'Δtat' is the complete *tat* deletion strain DADE-A. (**C**) Membranes from the same strains were isolated and immunoblotted with a combination of TatB and TatC antibodies or with TatA antibodies.

The following figure supplement is available for figure 5:

**Figure supplement 1.** Analysis of Tat transport activity.

concentration of the TatB E8A variant in cell membranes is lower than that of the wild-type TatB protein (*Figure 5C*; strain AE pB^E8AC), this cannot account for the complete loss of Tat function seen for the variant. Thus, removing the TMH polar residue of either TatA or TatB abolishes or severely compromises the transport of different Tat substrates.

We constructed a M205A/T208A/Q215A variant that contains substitutions of all three polar cluster residues in TatC and which is hereafter designated 'MTQ'. This variant was unable to support Tat transport (*Figure 5A,B*; strain A pBC^MTQ). To determine the relative contributions of the three substituted amino acids to Tat transport we assessed the transport activity of the corresponding single amino acid variants. All three variants were able to suppress cell chaining and permit CueO export, although the amount of CueO reaching the periplasm was lower than that observed with the wild-type protein particularly in the T208A and Q215A variants (*Figure 5A,B*; strains A pBC^M205A, A pBC^T208A, and A pBC^Q215A). The failure of the MTQ variant to support Tat transport is, therefore, due to more than one of the constituent substitutions. Analysis of doubly substituted TatC variants showed that the combination of T208A with Q215A is sufficient to abolish CueO export and exhibits partial cell chaining (*Figure 5A,B*; strain A pBC^T208A,Q215A). This suggests that it is the hydrogen bonding network in the polar cluster that is critical for transport. Immunoblotting experiments confirm that TatA, TatB, and TatC were still present in the membranes of each TatC polar cluster variant studied (*Figure 5C*).

In summary, removal of either the TatA/TatB or TatC sides of the predicted polar cluster prevents Tat transport, consistent with the idea that the polar cluster plays a crucial role in the Tat system.

## The polar cluster is required for TatBC interactions

Our structural model suggests that the polar cluster residues mediate complex formation between TatA/TatB and TatC. To test this idea we investigated the effect of polar cluster defects on protein-protein interactions within the *E. coli* Tat system.

In order to interpret these experiments it was first necessary to resolve a pre-existing uncertainty as to how TatA interacts with the TatBC complex. TatA is known to be transiently recruited to substrate-activated TatBC complexes. However, a small amount of TatA has also been reported to associate with the *E. coli* TatBC complex in the absence of substrate raising the possibility that TatA has a second mode of interaction with the TatBC complex (*Behrendt and Brüser, 2014*; *Bolhuis et al., 2001*; *De Leeuw et al., 2002*; *Zoufaly et al., 2012*). At the outset of this work, it was unclear whether the reported substrate-independent TatA binding was an authentic feature of the *E. coli* Tat system or an experimental artefact arising from the high level overproduction of Tat proteins in these studies. To resolve this uncertainty we analyzed the interactions between *E. coli* Tat proteins at native levels of expression. Membranes from wild-type *E. coli* cells were solubilized in digitonin, a detergent that is known to maintain the TatBC complex in an intact state (*Orriss et al., 2007*). Under these conditions TatA, as well as TatB, was found to co-immunoprecipitate with TatC (*Figure 6A*). We repeated the experiment using the 'FEA' variant of TatC (TatC^F94A,E103A) which blocks signal peptide binding and is therefore unable to undergo substrate-induced TatA oligomerization (*Alcock et al., 2013*; *Holzapfel et al., 2007*; *Rollauer et al., 2012*). The amount of TatA co-immunoprecipitated with the FEA variant was similar to that co-immunoprecipitating with the parental TatC protein (*Figure 6A*). This confirms that the native *E. coli* TatBC complex binds TatA molecules even in the absence of substrate activation. Thus, TatA has both constitutive and substrate-induced modes of interaction with the TatBC complex. The constitutively associating TatA molecules represent only a small proportion of the total TatA present in the cell (*Figure 6A*). Indeed, earlier studies with overproduced Tat proteins suggest that such constitutively associating TatA molecules are probably present at a equimolar ratio with TatC (*Bolhuis et al., 2001*; *Zoufaly et al., 2012*) in contrast to the 50 to 100-fold molar excess of TatA over TatC found in cells (*Berks et al., 2003*; *Jack et al., 2001*).

Co-immunoprecipitation experiments were used to assess the effects of polar cluster defects on the association of TatC with TatB and with constitutively bound TatA. These experiments were carried out on Tat proteins expressed at native levels and employed the FEA variant of TatC to exclude the possibility that any of the TatA molecules bound to TatC were part of substrate-induced TatA oligomers. The triple substitution variant MTQ was used to disrupt the polar cluster site on TatC. This change abolished the interaction of TatB with TatC but had no effect on the co-immunoprecipitation of constitutively associated TatA with TatC (*Figure 6B*, strain AE pBC^FEA, MTQ, compare lanes

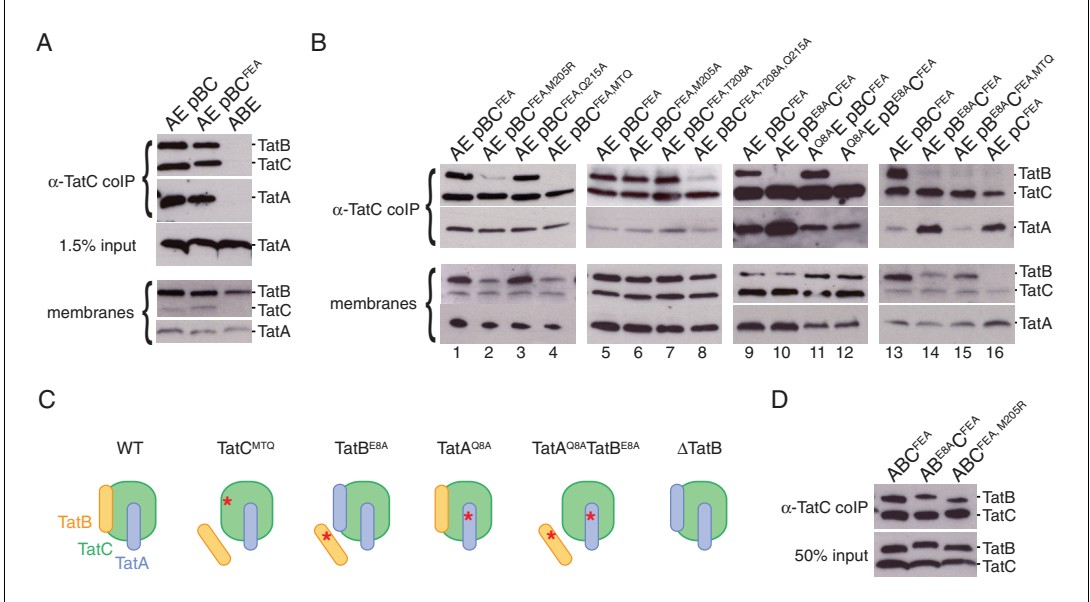

**Figure 6.** Polar cluster substitutions impair TatBC interactions. Digitonin-solubilized cell lysates of the indicated strains were immunoprecipitated with antibodies against TatC and then immunoblotted with either a combination of TatB and TatC antibodies or with TatA antibodies ('α-TatC coIP' panels). Where indicated a portion of the total cell lysate was also immunoblotted ('input' panels). (**A,B**) Co-immunoprecipitation analysis of strains expressing native levels of Tat proteins. Corresponding immunoblots of membranes isolated from the same strains are also shown ('membranes'). The TatC^FEA variant is blocked in substrate interactions. The TatC^MTQ variant carries the three polar cluster substitutions M205A, T208A, and Q215A. (**C**) Schematic representation of the results of polar cluster substitutions on Tat receptor complex composition. The red asterisks indicate the location of the polar cluster substitutions. (**D**) The indicated Tat proteins were overproduced approximately 500-fold in the ΔtatABCDΔtatE strain DADE-A from pUT2DH series plasmids and immunoprecipitated as in (**A**).

1 and 4). These observations suggest that the polar cluster site of TatC is occupied by TatB. They also show that constitutively associated TatA binds directly to TatC, rather than through TatB, and that this interaction does not require the polar cluster. We can discount the possibility that the loss of TatB interactions arises from improper folding of the TatC variant because the variant is still able to bind TatA.

If the polar cluster mediates TatB binding to TatC then the interaction between the two proteins should be abolished not only by eliminating the polar cluster site in TatC but also by removal of the TatB polar cluster residue E8. In agreement with this prediction, the E8A variant of TatB no longer co-immunoprecipitates with TatC (*Figure 6B*, compare lanes 9 and 10). By contrast, introducing the analogous Q8A substitution into TatA had no effect on substrate-independent binding of TatA to TatC (*Figure 6B*, compare lanes 9 and 11), consistent with the observed insensitivity of this interaction to removal of the TatC polar cluster (*Figure 6B*, lane 4). Taken together, the effects of substituting potential polar cluster residues in all three Tat components indicate that the polar cluster is involved in connecting TatB to TatC. This suggests that in the resting Tat system it is TatB that occupies the binding site located at TatC TM5/TM6.

MD simulations of the TatBC heterodimer model containing the polar cluster variants TatB E8A, TatC T208A, or TatC Q215A show that the periplasmic end of the TatB TMH is no longer tightly associated with TatC (*Figure 4A,B* and *Figure 4—figure supplement 1*) in agreement with the experimental data that the polar cluster plays an important role in binding TatB to TatC.

The obligatory role of the polar cluster in enabling TatB to bind to TatC would explain why the polar cluster is essential for Tat transport. Nevertheless, we observed that the TatC variants with single alanine substitutions within the polar cluster are unaffected in TatB binding (*Figure 6B*, compare lanes 1 and 3, 5 and 6, and 5 and 7) even though two of these variants are severely compromised in Tat transport activity (*Figure 5A*, strains AE pBC^T208A and AE pBC^Q215A). Thus, these additional

observations show that the polar cluster on TatC must have a mechanistic role in Tat transport beyond allowing complex formation with TatB.

An arginine substitution of TatC polar cluster residue M205 was previously isolated in a screen for Tat transport-deficient mutants (*Kneuper et al., 2012*). We confirmed the inability of this M205R variant to mediate Tat transport in the experimental system used in the current study (*Figure 5*; strain A pBC^M205R). Since alanine substitution of the same residue does not block Tat transport (*Figure 5A,B*; strain AE pBC^M205A) we deduce that it is the introduction of the arginine side chain, rather than loss of the methionine functionality, that prevents the M205R variant sustaining Tat transport. Co-immunoprecipitation experiments show that the TatC M205R substitution almost completely blocks complex formation with TatB, in contrast to the M205A substitution which has no effect on the binding of TatB to TatC (*Figure 6B*; compare lanes 2 and 6). The effects of the M205R substitution provide further support for the conclusion that TatB occupies the polar cluster site on TatC.

## TatA can replace TatB at the polar cluster site on TatC

Unexpectedly, the amount of TatA bound to TatC was markedly increased in the strain expressing the E8A variant of TatB (*Figure 6B*, compare lanes 9 and 10). Since the E8A substitution compromises the ability of TatB to bind to TatC, one possible explanation for this phenomenon is that removal of TatB from its binding site on TatC allows further TatA molecules to bind to TatC (*Figure 6C*). In agreement with this hypothesis we found that TatA binding to TatC was also increased in a strain lacking TatB (*Figure 6B*, compare lane 13 with lane 16). We hypothesized that the additional TatA molecules were binding to the site on TatC vacated by TatB. Consistent with this idea, no enhancement of TatA binding was observed when the TatB binding site was disrupted through introducing either the MTQ or M205R substitutions into TatC (*Figure 6B*, compare lane 1 with lanes 2 and 4). Our co-evolution-derived structural models suggest that TatA would bind to the TM5/TM6 site on TatC in a similar way to TatB, with TatA Q8 mediating the interaction with the TatC polar cluster (*Figure 4C*). We found that a TatA Q8A substitution was able to block the increase in TatA binding to TatC seen when TatB is absent (*Figure 6B*, compare lane 15 with lanes 14 and 16) providing strong evidence that the enhanced TatA binding phenomenon corresponds to TatA molecules binding to the TM5/TM6 site on TatC. MD simulations of the TatAC dimer model containing a Q8A substitution support the view that this amino acid change would weaken the interaction of a TatA molecule bound at the TM5/TM6 site in TatC (*Figure 4A,B* and *Figure 4—figure supplement 1*). *Figure 6C* summarizes our interpretation of the effects of polar cluster substitutions on Tat receptor complex composition.

## Investigating the effect of polar cluster substitutions on TatA oligomerization

We next assessed whether the polar cluster is involved in substrate-induced TatA oligomerization. We used a previously described experimental system in which the native TatA proteins have been replaced with a TatA-YFP fusion (*Alcock et al., 2013*). When the TatA-YFP fusion is in the dispersed state it is visualized as a halo of fluorescence at the periphery of the cell (*Figure 7A*, strain AyBCE, -CueO column). Overproduction of a substrate protein induces TatA oligomerization which results in TatA-YFP coalescing into bright mobile spots (*Figure 7A*, strain AyBCE, +CueO column). TatA-YFP oligomerization is reversed when the transmembrane proton-motive force (PMF) is collapsed by treatment with the protonophore carbonyl cyanide-*m*-chlorophenyl hydrazine (CCCP) leading to the re-appearance of the fluorescent halo (*Figure 7A*, strain AyBCE, +CCCP column).

The effects of TatC polar cluster substitutions on TatA oligomerization were entirely in accordance with their biochemical behavior (*Figure 7—figure supplement 1*). Variants which prevent TatB associating with TatC (strains AyE pBC^MTQ, AyE pBC^M205R) were unable to form TatA-YFP oligomers, consistent with the previously reported phenotype of strains lacking the TatB protein (*Leake et al., 2008*), whilst variants which retained TatBC interactions (strains AyE pBC^T208A, AyE pBC^Q215A) were still able to form protonophore-sensitive TatA-YFP oligomers.

We next investigated the effects of removing the predicted polar cluster residue Q8 from TatA. We first examined cells in which all other Tat components were present, including the TatA paralogue TatE. In these cells the oligomerization behavior of the TatA Q8A variant was

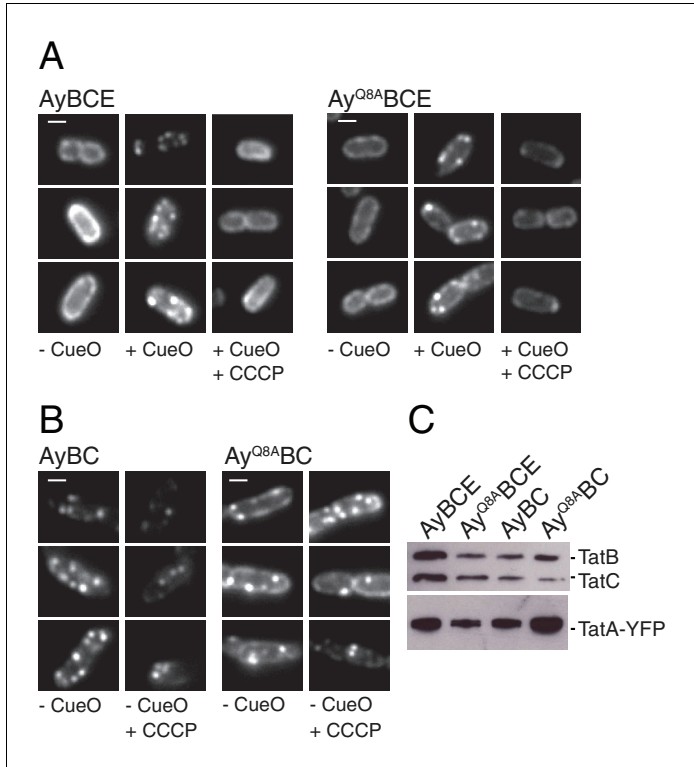

**Figure 7.** The TatA polar cluster residue is not required for TatA oligomerization. (**A,B**) Fluorescence images of TatA-YFP in living cells. The indicated strains were either left untreated (-CueO columns) or the Tat substrate protein CueO was overproduced from plasmid pQE80-CueO by adding 1 mM IPTG to early exponential phase cultures for 30 min prior to imaging (+CueO columns). 50 μM CCCP was subsequently added as indicated (+CCCP columns). Scale bar = 1 μM (**C**) Membranes isolated from the same strains were immunoblotted using a combination of TatB and TatC antibodies or with TatA antibodies to assess protein expression levels.

The following figure supplement is available for figure 7:

**Figure supplement 1.** The effect of TatC polar cluster substitutions on the substrate-induced assembly of TatA.

indistinguishable from that of the parental protein. In each case TatA-YFP oligomerization was induced by provision of substrate proteins and the resulting TatA-YFP complexes underwent disassembly following collapse of the PMF by addition of a protonophore (**Figure 7A**, compare strains AyBCE and Ay$^{Q8A}$BCE). These observations demonstrate that Q8 does not need to be present in a TatA protomer for that protein to be included within the TatA oligomer. Indeed, the TatA Q8A variant is able to increase the Tat pathway activity of the TatE-containing strain whether fused to YFP or not (**Figure 5** and **Figure 5—figure supplement 1B**). This shows that the variant TatA protein is incorporated into the Tat system in such a way that it can contribute to Tat function, albeit at a level considerably less than that supported by the wild-type TatA protein. We then examined the behavior of the TatA$^{Q8A}$-YFP fusion when TatE is absent. Under these conditions the wild-type TatA-YFP fusion is known to exhibit perturbed behavior. TatA-YFP oligomers are present even without substrate overproduction and these oligomers are insensitive to protonophore treatment (**Alcock et al., 2013**; **Leake et al., 2008**). The variant TatA$^{Q8A}$-YFP fusion again phenocopies the behavior of the parental fusion protein by forming substrate and protonophore-insensitive oligomers (**Figure 7B**, strains AyBC and Ay$^{Q8A}$BC). This provides further evidence that Q8 is not essential for TatA to be assembled into an oligomer and eliminates the possibility that the TatA$^{Q8A}$-YFP-containing oligomers are held together by TatE.

## The behavior of Tat variants is concentration-dependent

Some of the residues involved in the predicted polar cluster have been the subject of substitution analyses in previous studies of the *E. coli* Tat system. However, the reported effects of these substitutions are not always in agreement with the behavior observed here. For example, a TatB E8A variant is reported to be fully active (*Barrett and Robinson, 2005*; *Hicks et al., 2003*) while we find that this variant does not support Tat transport (*Figure 5A,B*, strain AE pB$^{E8A}$C). Similarly, whilst we observe that the TatC M205R substitution almost completely blocks complex formation between TatB and TatC (*Figure 6B*), the same variant has previously been reported not to affect TatBC complex assembly (*Kneuper et al., 2012*).

In the current work we have analyzed Tat components expressed at native levels whereas the earlier studies worked with overproduced Tat proteins. To investigate whether this variation in expression level could explain the discrepancy between our current observations and the earlier reports we reanalyzed the behavior of the TatB E8A and TatC M205R variants but now in strains overproducing all of TatA, TatB, and TatC. In contrast to the results obtained at native level expression (*Figure 6B*, lanes 2 and 10), overproduction of the variant Tat systems resulted in the co-immunoprecipitation of TatB with TatC (*Figure 6D*). These data show that mutagenic destabilization of the TatBC contact interface can be overcome by increasing the concentration of the interacting proteins and demonstrate that the observed behavior of Tat variants can be sensitive to their expression level.

## Evolutionary co-evolution analysis identifies additional inter-subunit contact sites within the TatBC complex

Our sequence co-evolution analysis using the TatA$_{All}$ dataset predicts that TatA family proteins interact not only with the C-terminus of TatC but also with TM1 of TatC via the contact pairs 18–21 and 11–24 (*Figure 1B,C*). The 18–21 contact is also detected with very high confidence (precision >0.8) in the TatB subset (*Figure 3—figure supplement 1A*). By contrast, contacts to TM1 are not seen with the TatA subset suggesting that the TM1 contact represents a TatB-specific interaction (*Figure 3—figure supplement 1A*). The distance between the characterized TatB binding site on TM5/TM6 and the predicted contacts on TM1 is too long to allow a TatB TMH to simultaneously interact with both sites on a single TatC molecule. However, the presence of these two binding sites could be accommodated by models for the organization of the TatBC complex in which each TatB TMH is sandwiched between two different TatC molecules. Contacts between TatB and TatC at two different sites have previously been invoked to explain TatBC crosslinking patterns (*Aldridge et al., 2014*; *Blümmel et al., 2015*; *Cline, 2015*; *Rollauer et al., 2012*; *Zoufaly et al., 2012*).

To experimentally test the predicted interaction site for TatB on TatC TM1 we attempted to disrupt the proposed interface by substituting bulky tryptophan residues at the most strongly predicted TatB V18-TatC L21 contact pair. CueO export was substantially decreased in the TatB V18W variant, and partially reduced in the TatC L21W variant, consistent with the perturbation of an important protein-protein interface (*Figure 8A*). Co-immunoprecipitation experiments were used to investigate whether the tryptophan substitutions affect TatBC interactions at native levels of expression (*Figure 8B*). Although the variant TatBC complexes remained intact following solubilization by digitonin, all of the tryptophan substitutions induced partial dissociation of TatB from TatC when solubilized using the detergent lauryl maltose neopentyl glycol (LMNG). The strongest effect was seen for the doubly substituted TatB V18W/TatC L21W variant. The disruptive effect of the tryptophan substitutions on TatBC interactions is consistent with the affected residues forming an inter-subunit contact site.

We used cysteine-scanning mutagenesis to probe for TatBC interactions in the vicinity of the proposed TM1 contact site. Cysteine substitutions at positions 21 to 25 in TatC were combined with either TatB L17C or V18C in a natively expressed Tat system. The cysteine-substituted pairs were then tested for inter-subunit disulfide crosslinking by addition of oxidant to live cells. Strong crosslinking was seen only for the TatB V18C-TatC L21C combination, which corresponds to the contact pair identified by the co-evolution analysis (*Figure 8C* and *Figure 8—figure supplement 1A*). Individually the TatB V18C and TatC L21C variants showed only moderate impairment of CueO export activity (*Figure 8—figure supplement 1B*). However, CueO transport was almost completely blocked by combining the two substitutions (*Figure 8—figure supplement 1B*) even though these two variants are not disulfide-linked under the conditions employed in our transport assay

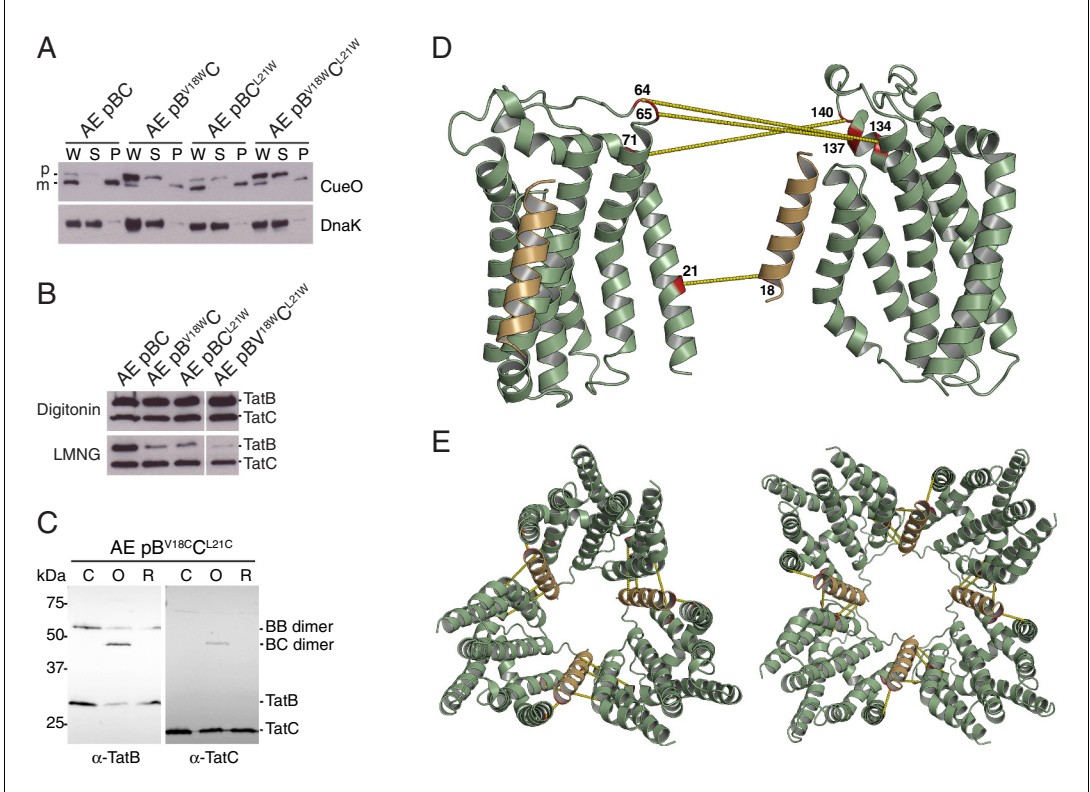

**Figure 8.** Identification of a second TatB_{TMH}-TatC contact site. (**A**) Tat transport activity of strains with tryptophan substitutions targeting the predicted interface between the TatB TMH and TatC TM1. Methodology and labels are as for *Figure 5A*. (**B**) Effects of the tryptophan substitutions on TatBC interactions. Cell lysates were solubilized in either digitonin (top panel) or LMNG (bottom panel), immunoprecipitated with antibodies against TatC, and then immunoblotted with a combination of TatB and TatC antibodies. (**C**) Disulfide crosslinks can be detected at the predicted interface between the TatB TMH and TatC TM1. Cells carrying the indicated cysteine substituted Tat variants were subjected to a mock incubation ('C', no oxidant or reductant), oxidizing ('O', copper phenanthroline) or reducing ('R', DTT) conditions. Membranes were then isolated and subjected to immunoblotting with TatB (left panels) or TatC (right panels) antibodies. (**D**) Structural representation of the highest-scoring co-evolution-predicted contacts between TatB_{TMH}C heterodimers (precision >0.6). (**E**) Model for the TatBC complex based on docking either three (Left) of four (Right) TatB_{TMH}-TatC heterodimers to optimize agreement with the co-evolution data in (**D**). The complexes are viewed from the cytoplasmic side of the membrane. See also *Figure 8—figure supplement 3*, *Video 3* and *Supplementary files 1* and *2*.

The following figure supplements are available for figure 8:

**Figure supplement 1.** TatB V18C and TatC L21C substitutions permit crosslinking of TatB to TatC and impair transport activity.

**Figure supplement 2.** Evolutionary contacts predicted by PSICOV for TatC.

**Figure supplement 3.** Model for the TatBC complex based on docking either (**A**) three or (**B**) four TatB_{TMH}-TatC heterodimers to optimize agreement with the co-evolution data in *Figure 8D*.

**Figure supplement 4.** Structural stability plots for the modeled trimeric and tetrameric Tat protein complexes from molecular simulations.

(*Figure 8—figure supplement 1C*). Taken together, the amino acid substitution studies and disulfide crosslinking analysis strongly support the prediction of a TatB V18-TatC L21 interaction.

We identified further potential protein-protein contacts within the TatBC multimer by selecting high-scoring TatC-TatC evolutionary couplings that do not match the fold of the isolated TatC protein. These pairs represent probable contacts between TatC subunits (*Figure 8—figure supplement 2*). Three of these pairs have a precision >0.6 and are located in the periplasmic cap of TatC (*Figure 8—figure supplement 2B*). The positions of these contacts are consistent with recent cross-

linking and genetic data which have suggested that the periplasmic cap mediates TatC self interactions (*Blümmel et al., 2015*; *Cléon et al., 2015*; *Ma and Cline, 2013*; *Zoufaly et al., 2012*).

We built models for the oligomeric TatBC complex by docking together multiple copies of the TatB$_{TMH}$C heterodimer model derived earlier (*Figure 3*) using the inferred inter-heterodimer contact pairs shown in *Figure 8D* as unambiguous restraints in the program Haddock with symmetry applied. Because the number of heterodimers in the TatBC oligomer is not firmly established, models containing either three or four copies of the TatB$_{TMH}$C heterodimer were produced. The final models are shown in *Figure 8E*, *Figure 8—figure supplement 3*, and *Video 3*, and the corresponding PDB format structure files are provided as *Supplementary files 1* and *2*. The TatB APH was not modeled as we do not have reliable evolutionary couplings or high-resolution experimental data to constrain the position of this part of the TatB structure.

The TatBC complex models show a hollow dome-shaped structure in which the concave surfaces of the TatC molecules face the interior of the particle (*Figure 8E*). In this arrangement the evolutionary coupled intermolecular TatC-TatC residue pairs would be able to interlock adjacent TatC molecules with limited rearrangements of the most flexible regions of the periplasmic cap loops (*Figure 8—figure supplement 3* and *Video 3*).

The modeled TatBC complexes were subjected to atomistic MD simulations in a membrane environment (*Figure 8—figure supplement 4* and *Video 4*). Simulations were run with the central cavity filled either with phospholipids, to reflect the fact that the proteins are inserted into a membrane bilayer, or with water, to examine the possibility that the cavity could form part of a transmembrane conduit for Tat substrates. The models were stable over the 100 ns simulation time in both scenarios suggesting significant structural cohesion within the protein complex and an ability to tolerate molecules of different polarities within the central cavity. Nevertheless, the simulations with the water-filled cavity showed consistently higher mobility than the simulations with a lipid-filled cavity raising the possibility that over a longer time span only the lipid-supported cavity is stable. Intuitively, it seems unlikely that TatBC would permanently contain a water-filled pore since this would result in the leakage of protons and other ions through the Tat site.

## Discussion

In this work we have used evolutionary and experimental data to obtain molecular level insight into how the TatA paralogs TatA and TatB interact with the other essential Tat pathway component TatC. At the outset of this work it was thought that TatA and TatB engaged in completely distinct patterns of protein-protein interaction: TatB was considered to form a permanent complex with TatC to bind substrate proteins, whilst TatA was specifically recruited to the substrate-bound state of the TatBC complex in stoichiometric excess over the other Tat components. Unexpectedly, we have uncovered similarities in the way the two TatA paralogs interact with TatC.

For TatB we have defined the interactions with TatC in considerable detail and have used this information to produce a well-validated, residue-level model for the structure of the *E. coli* TatBC substrate receptor complex (*Figure 8E*). The model involves a circle of alternating TatB and TatC proteins in which the TMH of each TatB molecule is sandwiched between the opposite ends of two TatC molecules. One face of the TatB TMH has extensive contacts with a site formed by TM5 and TM6 of TatC. This interface includes hydrogen-bonding interactions between the polar residue found in the TMH of TatB and two polar residues on TatC. Our experimental analysis shows that this intramembrane polar cluster is essential for Tat function. The opposite face of the TatB TMH forms a more limited interaction interface with TM1 of a second TatC molecule.

For TatA, our key and unanticipated finding is that this protein interacts with the same TM5/TM6 site on TatC that is used to bind TatB. The evidence for this deduction is as follows. Firstly, TatA conserves the TMH polar residue that is

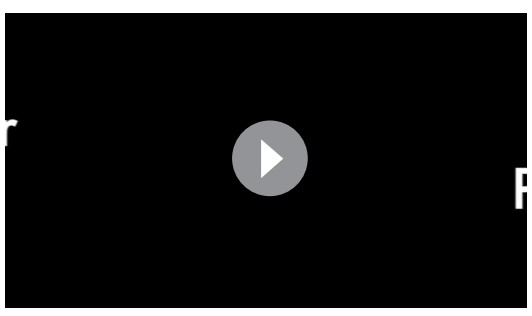

**Video 3.** Animations of the (TatBC)$_3$ and (TatBC)$_4$ complex models.

used by TatB for polar cluster formation at the TM5/TM6 site, and this residue is functionally important in TatA proteins (*Figure 5A,B*)(*Dabney-Smith et al., 2003*; *Greene et al., 2007*). Secondly, the TM5/TM6 contact site on TatC is predicted by co-evolution analysis when either TatA or TatB datasets are used (*Figure 3* and *Figure 3—figure supplement 1*). Thirdly, molecular models with either TatA or TatB docked at the TM5/TM6 site are stable in MD simulations indicating that both TatA and TatB are structurally compatible with this binding site (*Figure 4A* and *Figure 4—figure supplement 1A,C*, *Video 1*). Importantly, in both the TatAC and TatBC simulations, the TMH polar residue is involved in polar cluster formation and the complexes are perturbed in the simulations when the TMH polar residue is removed from either TatA or TatB (*Figure 4A,B* and *Figure 4—figure supplement 1B,D*, *Video 2*). Fourthly, we find that *E. coli* TatA is able to bind to the TM5/TM6 site if the site is vacated by TatB. This deduction arises from the observations that increased levels of TatA associate with TatC when TatB is absent, but that this increase is blocked by substitutions at either the TatA TMH polar residue or the TatC polar cluster (*Figure 6B*). Whilst this experiment involves a non-physiological manipulation of the Tat system, it nevertheless demonstrates that TatA is capable of sequence-specific binding at the TM5/TM6 site. Significantly, this TM5/TM6 contact is the only mode of protein-protein interaction involving TatA that depends on the TMH polar residue. Specifically, neither a substrate-independent TatAC interaction identified in this work (*Figure 6A*), nor formation of TatA oligomers in *E. coli* requires the TMH polar residue (*Figures 6B* and *7A,B*). Since the TatA TMH polar residue is important for Tat transport (above), this implies that the observed interaction of TatA at the TatB binding site is mechanistically relevant.

The proposal that both TatA and TatB are able to use the same binding site on TatC is supported by the observation that the TatA and TatB proteins of plant chloroplasts have biochemical behavior that closely mirrors that of their *E. coli* counterparts (*Cline, 2015*), even though they are very similar in sequence to each other (*Figure 2A*, red star). The resemblance between the two chloroplast TatA paralogs is so strong, including identical TMH polar residues, that it is difficult to imagine that only one is capable of interacting with the TM5/TM6 binding site on TatC. The high similarity of the chloroplast TatA and TatB proteins implies that they arose from a recent gene duplication event (*Berks et al., 2003*) and suggests that modification of TatA to have a TatB function is structurally subtle and has arisen independently on more than one occasion.

Taken together, our data provide strong evidence that TatA and TatB share a binding site on TatC. Since TatB occupies this site in the resting TatBC complex, it follows that TatA must displace TatB from the site at some stage in the translocation cycle. Although it might seem unlikely that the TatBC interface could be disrupted to allow this to happen, we have shown that ablation of the polar cluster is sufficient to release TatB from TatC in detergent solution (*Figure 6B*) indicating that the helix-helix packing interaction between the two proteins is inherently labile. Substrate binding to the TatBC complex is known to produce an organizational change in the Tat system and so it is highly likely that signal peptide docking is the trigger for TatA for TatB exchange. In our TatBC oligomer model, TM1 of TatC is in contact with both the signal peptide of the substrate molecule and the TMH of TatB providing an obvious way to physically link substrate binding to movement of TatB. Thus, conformational change at TM1 elicited by signal peptide binding can be envisaged to reposition the adjacent TatB TMH, thereby reducing its affinity for the TM5/TM6 site on the neighboring TatC molecule.

The proposal that substrate binding is mechanistically linked to displacement of TatB from the TM5/TM6 site is able to explain the otherwise puzzling earlier genetic observations that certain substitutions at or near the TatB TMH polar residue (*E. coli* TatB E8K or F9Q variants) permit the export of substrate proteins with defective signal peptides even though the substitutions do not fall within the signal peptide binding site of TatC (*Kreutzenbeck et al., 2007*; *Lausberg et al., 2012*). Interpreting the effects of these substitutions in the light of the structural data presented here suggests that the substitutions weaken the interaction between the TatB TMH and TatC TM5/TM6. Importantly, this is also what our mechanistic model proposes has to occur upon signal peptide binding. We, therefore, deduce that the TatB substitutions work by reducing the activation barrier to TatB displacement so that even relatively weakly binding signal peptides are able to trigger the structural transformations leading to TatA uptake. Notably, these TatB substitutions would not affect the subsequent, mechanistically essential, interaction of TatA with TatC at the same site. Indeed, it is probably significant that equivalent substitutions at this binding site were not identified in TatA or TatC (*Kreutzenbeck et al., 2007*; *Lausberg et al., 2012*).

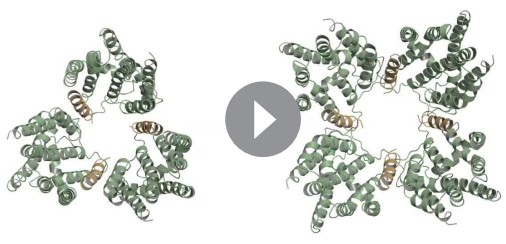

**Video 4.** Exemplar atomistic 100 ns molecular simulations of the (TatBC)$_3$ (Left) and (TatBC)$_4$ (Right) complexes in a membrane bilayer. The complexes are viewed from the cytoplasmic side of the membrane. The simulations were run with phospholipids filling the internal pore of the complex.

The data presented here confirm that *E. coli* TatA has an additional, substrate-independent, mode of interaction with TatC (*Figure 6A*). The TatAC contact site involved in this interaction is clearly distinct from the TM5/TM6 site because the interaction is insensitive to polar cluster substitutions on either component or to the presence of TatB. Nevertheless, the two sites are likely to be closely adjacent given the very similar site-specific crosslinking patterns to TatC exhibited by TatB and TatA under non-transport conditions (*Aldridge et al., 2014*; *Zoufaly et al., 2012*). This additional TatA binding site could function to position a TatA molecule in readiness to occupy the TatB binding site.

In summary, we infer that TatA and TatB compete for a single binding site on TatC. In the resting Tat system TatB has the higher affinity for the site, while in the substrate-activated state TatA binding is favored. We propose that the TatA molecule that is recruited to this shared binding site nucleates the uptake of further TatA molecules to form the transport-capable translocation complex (*Figure 9*).

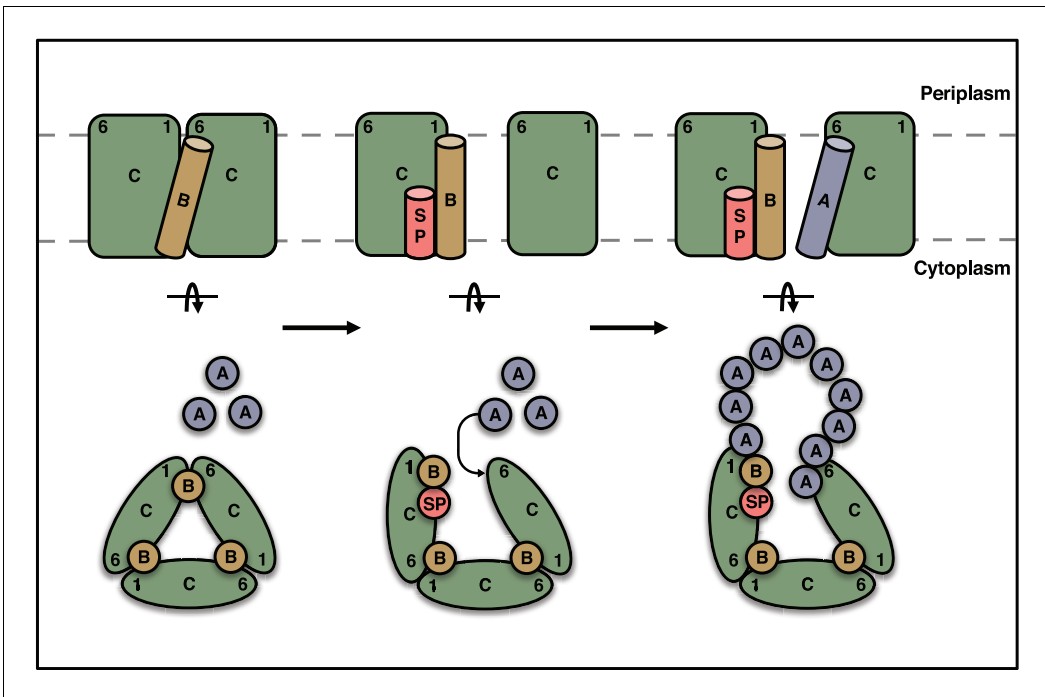

**Figure 9.** Schematic model for substrate activation of the Tat receptor complex. The TMHs of TatB molecules (orange) in the receptor complex are sandwiched between TM1 of one TatC molecule and TM6 of the adjacent TatC molecule (green). Signal peptide (SP, red) binding to a TatC subunit transmits a conformational change through TM1 that reduces the affinity of TatB for the TM6 site on the adjacent TatC molecule and favors uptake of TatA (blue) into this site. The incoming TatA molecule is envisaged to nucleate the formation of the TatA oligomer. Current data are insufficient to determine the final location of the displaced TatB molecule and so the position shown should be regarded as speculative. A (TatBC)$_3$ oligomer is shown for simplicity, but this mechanism is generally applicable to a (TatBC)$_n$ complex.

## Materials and methods

### Sequence co-evolution analysis

Amino acid sequences for the Tat subunits were downloaded from UniProt (*UniProt Consortium, 2015*). Multiple sequence alignments (MSAs) were generated separately for TatC and the particular TatA family dataset of interest using Clustal Omega (*Sievers et al., 2011*). To eliminate irrelevant sequences from the alignments, all TatA family sequences lacking the invariant inter-helix hinge residue G21, or TatC family sequence lacking essential signal peptide binding residue E103, were removed from the alignments. Each entry in the TatA MSA was then concatenated to each entry in the TatC MSA bearing the same organism ID and the resulting MSA filtered to retain only non-identical sequences. Any TatA sequence with no TatC partner in the same organism was removed from the analysis. PSICOV (*Jones et al., 2012*) was used to predict the co-evolving residue pairs shown in the main text. Evolutionary couplings calculated using the alternative programs CCMPRED (*Seemayer et al., 2014*), FreeContact (*Kaján et al., 2014*), and MetaPSICOV (*Jones et al., 2015*) resulted in similar contact predictions (*Table 1*). Topologically implausible contacts were removed by retaining only contacts that were less than 15 Å apart along the membrane normal, where the position of the subunits in the membrane was taken as the time-averaged location of the protein in coarse grain MD simulations in a membrane environment (*Figure 1—figure supplement 1*).

To identify TatA subsets, Clustal Omega was used to produce a MSA of all TatA sequences which was then input into ClustalW2 (*Larkin et al., 2007*) to produce a phylogenetic tree. The resulting tree was rendered as a cladogram using Figtree v1.4.2 (http://tree.bio.ed.ac.uk/software/figtree/).

### Molecular modeling and molecular simulations

Individual protein components were configured and built using Modeller (*Sali and Blundell, 1993*). $(TatB_{TMH}C)_3$ and $(TatB_{TMH}C)_4$ heteromers were built with, respectively, either C3 or C4 symmetry applied, by using both inter-TatC and inter-TatB-TatC residue pairs as unambiguous constraints in Haddock dockings (*Dominguez et al., 2003*).

All MD simulations were performed using GROMACS v5.0.2 (*Pronk et al., 2013*). The MemProtMD pipeline (*Stansfeld et al., 2015*) was used with the Martini 2.2 force field (*de Jong et al., 2013*) to run an initial 1 µs Coarse Grained (CG) MD simulation to permit the assembly and equilibration of a 1-palmitoly, 2-oleoyl, phosphatidylglycerol (POPG): 1-palmitoly, 2-oleoyl, phosphatidylethanolamine (POPE) bilayers, at a 1:4 ratio, around the Tat complexes. The final snapshot of the CGMD simulation was converted back to atomic detail (*Stansfeld and Sansom, 2011*) and any steric clashes between protein and lipids removed using Alchembed (*Jefferys et al., 2015*). Atomistic coordinates were initially equilibrated for one ns with position restraints placed upon the protein structure. The complexes were then subjected to a 100 ns MD simulation with position restraints lifted. Three repeat simulations were performed for each starting configuration. For the atomic simulations, the Gromos53a6 force field was used (*Oostenbrink et al., 2004*). Systems were neutralized with a 150 mM concentration of NaCl. All simulations were performed at 37°C, with protein, lipids, and solvent separately coupled to an external bath using the velocity-rescale thermostat (*Bussi et al., 2007*). Pressure was maintained at 1 bar with a semi-isotropic compressibility of $4 \times 10^{-5}$ using the Parrinello-Rahman barostat (*Parrinello and Rahman, 1981*). All bonds were constrained with the P-LINCS algorithm (*Hess, 2008*). Electrostatics was measured using the Particle Mesh Ewald (PME) method (*Darden et al., 1993*), while a Verlet cut-off scheme was employed to permit GPU calculation of non-bonded contacts,using Lennard-Jones parameters. Simulations were performed with an integration timestep of 2 fs.

MD simulations were analyzed using GROMACS tools (*Hess et al., 2008*), MDAnalysis (*Michaud-Agrawal et al., 2011*), and locally written code. All images and animations were generated using Pymol (*DeLano, 2002*).

### Strain and plasmid construction

Strains used in this work are listed in *Table 2*. For $A^{Q8A}$ strains, a *gln8ala* mutation was introduced into *tatA* in plasmid pKSUniA (*Koch et al., 2012*) by site-directed mutagenesis. An EcoRI-BamHI fragment encompassing $P_{tatA}tatA^{Q8A}$ was then subcloned into the shuttle vector pRS552

**Table 2.** Strains used in this study.

| Strain name* | Abbreviation | Genotype | Reference |
|---|---|---|---|
| MC4100 | ABCE | F⁻, ΔlacU169, araD139, rpsL150, relA1, ptsF, rbsR, flbB5301 | (*Casadaban and Cohen, 1979*) |
| MC4100-A | ABCE | arabinose-resistant derivative of MC4100 | (*Ize et al., 2002*) |
| J1M1 | ABC | MC4100 ΔtatE | (*Sargent et al., 1998*) |
| ELV16 | BCE | MC4100 ΔtatA | (*Sargent et al., 1998*) |
| B1LK0 | ABE | MC4100 ΔtatC | (*Bogsch et al., 1998*) |
| MΔBC | AE | MC4100 ΔtatBC | (*Alcock et al., 2013*) |
| DADE-A | Δtat | MC4100 ΔtatABC ΔtatE | (*Ize et al., 2002*) |
| MΔABC-A λAry | AyE | MC4100-A ΔtatABC::apra, attB::P$_{tatA}$tatA-EAK-eyfp$^{A206K}$(kanʳ) | (*Alcock et al., 2013*) |
| JARV16 λA(Q8A) | A$^{Q8A}$BC | MC4100 ΔtatA ΔtatE, attB::P$_{tatA}$tatA$^{Q8A}$ (kanʳ) | This work |
| MΔABC-A λA(Q8A) | A$^{Q8A}$E | MC4100-A ΔtatABC::apra, attB::P$_{tatA}$tatA$^{Q8A}$(kanʳ) | This work |
| ELV16 λA(Q8A) | A$^{Q8A}$BCE | MC4100 ΔtatA, attB::P$_{tatA}$tatA$^{Q8A}$ (kanʳ) | This work |
| ELV16 λAry(Q8A) | Ay$^{Q8A}$BCE | MC4100 ΔtatA, attB::P$_{tatA}$tatA$^{Q8A}$-EAK-eyfp$^{A206K}$ (kanʳ) | This work |
| ELV16-A λAry | AyBCE | MC4100-A ΔtatA, attB::P$_{tatA}$tatA-EAK-eyfp$^{A206K}$ (kanʳ) | (*Alcock et al., 2013*) |
| JARV16 λAry | AyBC | MC4100 ΔtatA ΔtatE, attB::P$_{tatA}$tatA-EAK-eyfp$^{A206K}$(kanʳ) | (*Leake et al., 2008*) |
| JARV16 λAry(Q8A) | Ay$^{Q8A}$BC | MC4100 ΔtatA ΔtatE, attB::P$_{tatA}$tatA$^{Q8A}$-EAK-eyfp$^{A206K}$(kanʳ) | This work |

*All strains designated '–A' are arabinose-resistant derivatives (*Ize et al., 2002*)

(*Simons et al., 1987*) and delivered onto the chromosome of the desired background strain at the *E. coli* phage lambda attachment site (*att*).

Plasmids used in this work are listed in *Table 3*. All codon changes were introduced by site-directed mutagenesis using the Quikchange method (Stratagene, San Diego, California). To allow co-ordinate overproduction of untagged TatA, TatB, and TatC, a stop-codon was introduced downstream of the *tatC* ORF in plasmid pUnitat2 (*McDevitt et al., 2005*) to form plasmid pUT2DH.

To achieve approximately wild-type levels of tatC expression, plasmid p101C*TatC was constructed as follows. The *tatC* gene was amplified from plasmid p101C*BC (*Alcock et al., 2013*) using primers BamHITatCF and SphITatCR (*Alcock et al., 2013*). The amplicon was digested with BamHI and SphI, then cloned into the same sites of p101CTatBC to give p101CTatC. This construct places *tatC* immediately downstream of the *tatA* promoter. To reduce the level of *tatC* expression from this plasmid, site-directed mutagenesis was used to simultaneously alter the ribosome-binding site, and change the *tatC* start codon to GTG, using primers TatRBS_TatC*_F (5′-CATCTACCACAGAG-CAGGATCCGTGTCTGTAGAAGATAC-3′) and TatRBS_TatC*_R (5′-GTATCTTCTACAGACACGGA TCCTGCTCTGTGGTAGATG-3′). Plasmid p101C*TatC$^{FEA}$ was then produced from p101C*TatC by two rounds of site-directed mutagenesis using the primer pairs TatCF94AF/TatCF94AR and Tat-CE103AF/TatCE103AR (*Alcock et al., 2013*).

For cysteine crosslinking, p101C*TatBC was modified to remove the four *tatC* cysteine codons as follows: *tatBC* was removed from p101C*TatBC by digestion with BamHI and SphI. *tatBC* lacking cysteine codons was amplified from pTat101 cys less (*Cléon et al., 2015*) with primers BamHI-TatB-F and SphI-TatC-R (*Alcock et al., 2013*), and digested with BamHI and SphI. The two fragments were ligated to give p101C*BC cys less.

## Analytical methods

For co-immunoprecipitation experiments, cultures of freshly transformed cells were harvested at mid-log phase and resuspended in IP buffer (10 mM Tris-HCl pH 7.6, 140 mM NaCl, 1 mM EDTA) containing 1 mM phenylmethylsulfonyl fluoride 0.1 mg/ml lysozyme and 0.1 mg/ml DNase I. For experiments using pUT2DH derivatives, protein expression was induced for 1 hr with 1 mM IPTG prior to harvesting. Cells were disrupted by sonication, cleared of debris by centrifugation for 3 min at 15,000x*g*, then solubilized with 1.5% digitonin (Calbiochem, San Diego, California) for 1 hr at 4°C. Cells were pre-cleared by incubating with 100 µl of a 50% agarose slurry during the solubilization

**Table 3.** Plasmids used in this study.

| Plasmid name | Abbreviation | Description | Reference |
|---|---|---|---|
| pTH19cr | | Low copy number pSC101-derived replicon. Chl$^r$. | (*Hashimoto-Gotoh et al., 2000*) |
| p101C*TatBC | pBC | pTH19cr derivative. Expression of *tatBC* from the *tatA* promoter with a modified RBS. | (*Alcock et al., 2013*) |
| p101C*BC M205R | pBC$^{M205R}$ | p101C*TatBC *tatC-met205arg* | This work |
| p101C*BC Q215A | pBC$^{Q215A}$ | p101C*TatBC *tatC-gln215ala* | This work |
| p101C*BC MTQ | pBC$^{MTQ}$ | p101C*TatBC *tatC- met205ala- thr208ala- gln215ala* | This work |
| p101C*BC M205A | pBC$^{M205A}$ | p101C*TatBC *tatC- met205ala* | This work |
| p101C*BC T208A | pBC$^{T208A}$ | p101C*TatBC *tatC- thr208ala* | This work |
| p101C*BC TQA | pBC$^{T208A,Q215A}$ | p101C*TatBC *tatC- thr208ala- gln215ala* | This work |
| p101C*BC FEA | pBC$^{FEA}$ | p101C*TatBC *tatC- Phe 94ala- glu103ala* | (*Alcock et al., 2013*) |
| p101C*BC FEAMR | pBC$^{FEA, M205R}$ | p101C*BC FEA *tatC-met205arg* | This work |
| p101C*BC FEAQA | pBC$^{FEA, Q215A}$ | p101C*BC FEA *tatC-gln215ala* | This work |
| p101C*BC FEAMTQ | pBC$^{FEA, MTQ}$ | p101C*BC FEA *tatC- met205ala- thr208ala- gln215ala* | This work |
| p101C*BC FEAMA | pBC$^{FEA,M205A}$ | p101C*BC FEA *tatC- met205ala* | This work |
| p101C*BC FEATA | pBC$^{FEA,T208A}$ | p101C*BC FEA *tatC- thr208ala* | This work |
| p101C*BC FEATQA | pBC$^{FEA,T208A, Q215A}$ | p101C*BC FEA *tatC- thr208ala- gln215ala* | This work |
| p101C*BC E8A | pB$^{E8A}$C | p101C*TatBC *tatB-glu8ala* | This work |
| p101C*BC EAFEA | pB$^{E8A}$C$^{FEA}$ | p101C*BC FEA *tatB-glu8ala* | This work |
| p101C*BC EFM | pB$^{E8A}$C$^{FEA, MTQ}$ | p101C*BC FEAMTQ *tatB-glu8ala* | This work |
| p101CTatC | | pTH19cr derivative expressing *tatC* from the *tatA* promoter | This work |
| p101C*TatC | pC | p101CTatC with a modified RBS and GTG start codon. | This work |
| p101C*TatC FEA | pC$^{FEA}$ | p101C*TatC *tatC- Phe 94ala- glu103ala* | This work |
| p101C*BC V18W | pB$^{V18W}$C | p101C*TatBC *tatB-val18trp* | This work |
| p101C*BC L21W | pBC$^{L21W}$ | p101C*TatBC *tatC-leu21trp* | This work |
| p101C*BC VLW | pB$^{V18W}$C$^{L21W}$ | p101C*TatBC V18W *tatC-leu21trp* | This work |
| pTat101 cys less | | Very low copy number vector expressing *tatABC*$^{C23A,C33A,C179A,C224A}$ from the *tat* promoter. Kan$^r$. | (*Cléon et al., 2015*) |
| p101C*BC cys less | pBC cys$^-$ | p101C*TatBC *tatC-cys23ala-cys33ala-cys179ala-cys224ala* | This work |
| p101C*BC V18C | pB$^{V18C}$C | p101C*BC cys less *tatB-val18cys* | This work |
| p101C*BC L21C | pBC$^{L21C}$ | p101C*BC cys less *tatC-leu21cys* | This work |
| p101C*BC 17C 21C | pB$^{L17C}$C$^{L21C}$ | p101C*BC L21C *tatB-leu17cys* | This work |
| p101C*BC 18C 21C | pB$^{V18C}$C$^{L21C}$ | p101C*BC L21C *tatB-val18cys* | This work |
| p101C*BC 17C 22C | pB$^{L17C}$C$^{N22C}$ | p101C*BC cys less *tatB-leu17cys tatC-asn22cys* | This work |
| p101C*BC 18C 22C | pB$^{V18C}$C$^{N22C}$ | p101C*BC V18C *tatC-asn22C* | This work |

*Table 3 continued on next page*

*Table 3 continued*

| Plasmid name | Abbreviation | Description | Reference |
|---|---|---|---|
| p101C*BC 17C 23C | pB$^{L17C}$C$^{C23C}$ | p101C*BC cys less *tatB-leu17cys tatC-ala23cys* | This work |
| p101C*BC 18C 23C | pB$^{V18C}$C$^{C23C}$ | p101C*BC V18C *tatC-ala23cys* | This work |
| p101C*BC 17C 24C | pB$^{L17C}$C$^{I24C}$ | p101C*BC cys less *tatB-leu17cys tatC-ile24cys* | This work |
| p101C*BC 18C 24C | pB$^{V18C}$C$^{I24C}$ | p101C*BC V18C *tatC-ile24cys* | This work |
| p101C*BC 17C 25C | pB$^{L17C}$C$^{I25C}$ | p101C*BC cys less *tatB-leu17cys tatC-ile25cys* | This work |
| p101C*BC 18C 25C | pB$^{V18C}$C$^{I25C}$ | p101C*BC V18C *tatC-ile25cys* | This work |
| pUnitat2 | | Expression of *tatABC$_{his}$* under the control of a T5 promoter | (*McDevitt et al., 2005*) |
| pUT2DH | | Expression of *tatABC* under the control of a T5 promoter. | This work |
| pUT2DH FEA | | pUT2DH *tatC- Phe 94ala- glu103ala* | This work |
| pUT2DH FEAMR | | pUT2DH FEA *tatC-met205arg* | This work |
| pUT2DH EAFEA | | pUT2DH FEA *tatB-glu8ala* | This work |
| pQE80-CueO | | Synthesis of *E. coli* CueO with a C-terminal his$_6$ tag. | (*Leake et al., 2008*) |
| pKSUniA | | *tatA* under control of the *tat* promoter in pBluescript KS$^+$ | (*Koch et al., 2012*) |
| pKSUniA Q8A | | pKSUniA *tatA-gln8ala* | This work |
| pRS552 | | lambda attachment site shuttle vector | (*Simons et al., 1987*) |
| p552TatA Q8A | | pRS552 carrying P$_{tatA}$tatA$^{Q8A}$ | This work |

step. After solubilization, samples were centrifuged for 1 hr at 100,000xg at 4°C. A portion of the supernatant was removed for SDS-PAGE ('input'), and the remainder was incubated with α-TatC antibodies for 1.5 hr at 4°C. 20 µl of a 50% slurry of Protein A-sepharose (Sigma-Aldrich, St. Louis, Missouri) was added, and incubation was continued for a further 1.5 hr at 4°C. Unbound material was removed and the Protein A-sepharose was washed by centrifugation with 2 × 1 ml IP buffer containing 0.1% digitonin. Bound proteins were then eluted in Laemmli sample buffer (*Laemmli, 1970*) for 10 min at 55°C. Samples were analyzed by SDS PAGE and immunoblotting.

Polyclonal antibodies against TatA, TatB, and TatC were raised in rabbits by Davids Biotechnologie, Regensburg, Germany (TatA, TatB) or Genscript, Nanjing, China (TatC). TatA antibodies were raised against the C-terminal domain of the TatA protein expressed from plasmid pFAT587 and purified as described by (*De Leeuw et al., 2001*). TatB antibodies were raised against a mixture of two peptides, encompassing residues 69–84 and 156–171. TatC antibodies were raised against a C-terminal peptide encompassing residues 238–258. Immunoblotting data are representative of experiments carried out a minimum of three times with independent biological replicas.

In vivo disulfide cross-linking experiments were performed using strain MΔBC with TatB and TatC variants produced from derivatives of plasmid p101C*TatBC cys less. Early exponential phase cultures were treated for 1 min with oxidant (1.8 mM copper phenanthroline), or reductant (10 mM DTT), or mock treated. Free sulfhydryls were then blocked by treatment with 8 mM N-Ethylmaleimide, 12 mM Na$_2$EDTA.

Cells were prepared for fluorescence microscopy and imaged as previously described by Alcock and co-workers (*Alcock et al., 2013*) for *Figure 7*, or Cleon and co-workers (*Cléon et al., 2015*) for *Figure 7—figure supplement 1*. Cell imaging panels show exemplar data from at least three independent cultures examined on different days.

## Acknowledgements

We thank Mark Sansom, Tim Nugent and David Jones for helpful discussions.

## Additional information

### Funding

| Funder | Grant reference number | Author |
|---|---|---|
| Biotechnology and Biological Sciences Research Council | BB/L002531/1 | Tracy Palmer<br>Ben C Berks |
| Wellcome | Investigator Award 107929/Z/15/Z | Ben C Berks |
| Medical Research Council | G1001640 | Tracy Palmer<br>Ben C Berks |
| European Commission | Marie Curie Fellowship Programme: GP7-PEOPLE-2013-IEF 626436 | Hajra Basit<br>Mark I Wallace |
| Biotechnology and Biological Sciences Research Council | BB/I019855/1 | Phillip J Stansfeld |
| Wellcome | Investigator Award 110183/Z/15/Z | Tracy Palmer |

The funders had no role in study design, data collection and interpretation, or the decision to submit the work for publication.

### Author contributions

FA, PJS, Conception and design, Acquisition of data, Analysis and interpretation of data, Drafting or revising the article; HB, JH, MABB, Acquisition of data, Analysis and interpretation of data; TP, Analysis and interpretation of data, Contributed unpublished essential data or reagents; MIW, Analysis and interpretation of data; BCB, Conception and design, Analysis and interpretation of data, Drafting or revising the article

### Author ORCIDs

Mark I Wallace, http://orcid.org/0000-0002-5692-8313
Ben C Berks, http://orcid.org/0000-0001-9685-4067

## Additional files

### Supplementary files

• Supplementary file 1. Molecular model for the $(TatBC)_3$ complex in PDB format.

• Supplementary file 2. Molecular model for the $(TatBC)_4$ complex in PDB format.

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
