## [Decision Letter]

Thank you for submitting your article "Assembling the Tat protein translocase" for consideration by *eLife*. Your article has been favorably evaluated by John Kuriyan (Senior Editor) and two reviewers, one of whom, [Nir Ben-Tal (Reviewer #1), is a member of our Board of Reviewing Editors. The following individual involved in review of your submission has agreed to reveal their identity: William M Clemons (Reviewer #2).

The reviewers have discussed the reviews with one another and the Reviewing Editor has drafted this decision to help you prepare a revised submission.

Summary:

The Tat protein complex mediates the transport of folded proteins across the cytoplasmic membrane in bacteria and the thylakoid membranes of plant chloroplasts. The complex is assembled from multiple copies of each of the following three proteins: TatA, TatB, and TatC. The manuscript describes the use of evolutionary data (mostly coevolution), molecular simulations, and various experimental assays to suggest a structure model of the interaction between the 3 subunits in the complex based on the structures of the individual components. It further suggests a model of Tat activation and conformational changes upon interaction with the signal peptide of the transported protein.

In particular, it explores a hypothesis generated from co-evolution analysis that TatA and TatB both bind to the conserved patch of hydrophilic residues on TM5 and TM6 of TatC. The work convincingly demonstrates that the paralogs TatA and TatB evolved from a common TatA family member. The divergence resulted in TatB occupying the binding state during the resting channel. Binding of substrate would displace TatB, resulting in TatA occupying this site. Additionally, TatA is demonstrated to have a second binding site in the resting translocase. The result is a reasonably convincing model of the TatB/C complex that is substantiated by molecular dynamics.

This is a very interesting and carefully conducted study, where predicted contacts between amino acid pairs are examined experimentally, e.g., using disulfide crosslinks. The results are nicely presented within the context of existing knowledge on Tat.

Essential revisions:

The model structures (in PDB format) should be made readily accessible to the public. Maybe as Supplementary data?

1) How dependent is the phylogenetic tree of Figure 2 on the method used to generate it? CLUSTAW might not always give the most reliable trees. More accurate tools, such as MAFFT-LINSI and PRANK should be used.

2) The charged residue in the central pocket isn't discussed at all. Based on the phylogeny determined, is there anything that can be gleaned from TatC homologs that only have a polar residue?

3) Introduction, fifth paragraph: 'direct methods' isn't a clear term. Perhaps 'standard structural methods'?

4) Subsection “Evolutionary contacts between TatA family proteins and TatC”, second paragraph: 'precision score' is used without context. Considering the general audience it might be helpful to provide some context for what the scores actually mean. Is 0.5 a reasonable cut-off? High/low. One can only infer from the figure but there should be a way to explain the score in a sentence or two.

5) Subsection “Exploring evolutionary contacts for different TatA paralogs”, second paragraph: some numbers would be interesting to cite. 'Almost all' means what? How prevalent are TatB?

6) Subsection “Exploring evolutionary contacts for different TatA paralogs”, end of second paragraph: what is meant by 'partners' in this context? Does it have to have a TatC?

7) Subsection “Exploring evolutionary contacts for different TatA paralogs”, fourth paragraph: TatA and TatA family are used interchangeably at times. It makes it unclear as TatA becomes defined. Here it should say 'TatA family subsets' for clarity.

8) Subsection “Exploring evolutionary contacts for different TatA paralogs”, fourth paragraph: perhaps change the two motifs to be F-G-X and X-G-P to make it easier to relate.

9) Subsection “Exploring evolutionary contacts for different TatA paralogs”, fourth paragraph: seems odd that E8 isn't discussed here.

10) Subsection “Evolutionary co-evolution analysis identifies additional inter-subunit contact sites within the TatBC complex”, last paragraph: can't really see the flexible regions of the periplasmic cap loops.

11) Subsection “Evolutionary co-evolution analysis identifies additional inter-subunit contact sites within the TatBC complex”, last paragraph: the rationale for filling with lipids or water should be discussed. Was anything learned here? It seems there should be some substantial differences. (I realize there are space constraints).

12) Figure 2: in the logo plot the length is constrained. How were differences in length accounted for in this presentation.

---

## [Author Response]

*Essential revisions:*

*The model structures (in PDB format) should be made readily accessible to the public. Maybe as Supplementary data?*

PDB files of the model structures were provided with the initial submission as [Supplementary-material SD1-data] and [Supplementary-material SD2-data] and these files are retained in the revised submission.

*1) How dependent is the phylogenetic tree of Figure 2 on the method used to generate it? CLUSTAW might not always give the most reliable trees. More accurate tools, such as MAFFT-LINSI and PRANK should be used.*

Clustal Omega, rather than ClustalW, was used to generate the multiple sequence alignment due to the inclusion of ~11,000 sequences. ClustalW was then used to create the phylogenetic tree from the Clustal Omega sequence alignment. We have clarified this in the text.

MAFFT-LINSI is limited to aligning 200 sequences and so could not be used to align the ~11,000 sequences employed in this work. PRANK also has difficulty with the alignment of this number of sequences, producing obviously sub-optimal sequence alignments. As we could not generate valid sequence alignments with these programs we are not able to produce phylogenetic trees for comparison with the ClustalW tree used in the paper.

*2) The charged residue in the central pocket isn't discussed at all. Based on the phylogeny determined, is there anything that can be gleaned from TatC homologs that only have a polar residue?*

The polar residue in the central pocket of TatC (Glu170 in the *E. coli* protein) is a prominent structural feature of this protein. It is predominantly conserved as either Glu (6404 sequences in our dataset) or Gln (2012 sequences), with only 329 sequences not having either residue at this position (in which case it is replaced with either Ala or Ile). There are no obvious patterns to the occurrence of Glu versus Gln at this position within a phylogenetic tree of TatC proteins. Additionally, this sequence position shows no significant evolutionary coupling with other amino acids in any of the Tat proteins. This is not surprising given the low level of variation at this position. Because our analysis of this structural feature provides no new insight into its function, and because nothing can be inferred from the absence of evolutionary couplings to this residue, we have not described this analysis in the text.

*3) Introduction, fifth paragraph: 'direct methods' isn't a clear term. Perhaps 'standard structural methods'?*

The text has been changed as suggested by the reviewers.

*4) Subsection “Evolutionary contacts between TatA family proteins and TatC”, second paragraph: 'precision score' is used without context. Considering the general audience it might be helpful to provide some context for what the scores actually mean. Is 0.5 a reasonable cut-off? High/low. One can only infer from the figure but there should be a way to explain the score in a sentence or two.*

The meaning of the term `precision score’ is now briefly set out at first occurrence in the text together with the rationale for the 0.5 cut-off.

*5) Subsection “Exploring evolutionary contacts for different TatA paralogs”, second paragraph: some numbers would be interesting to cite. 'Almost all' means what? How prevalent are TatB?*

The proportions of organisms within the analysed classes containing these combinations of TatA and TatB are now given.

*6) Subsection “Exploring evolutionary contacts for different TatA paralogs”, end of second paragraph: what is meant by 'partners' in this context? Does it have to have a TatC?*

We agree that this was confusingly worded and now clarify that `partners’ here means that the organism contains both a member of the TatA subset and a member of the TatB subset.

Organisms that do not contain both a TatA family protein and a TatC protein do not appear in our co-evolution analysis as they would not provide a linked TatA-TatC pair. This is a generic feature of the analysis method and we have added a sentence to the Methods section to reflect this. In reality almost all organism with a TatC protein also contain a TatA family protein with the very small minority that do not probably reflecting incompleteness in the source genome sequences.

*7) Subsection “Exploring evolutionary contacts for different TatA paralogs”, fourth paragraph: TatA and TatA family are used interchangeably at times. It makes it unclear as TatA becomes defined. Here it should say 'TatA family subsets' for clarity.*

We have made this suggested change. Elsewhere in the manuscript we have made equivalent alterations where this would clarify the text.

*8) Subsection “Exploring evolutionary contacts for different TatA paralogs”, fourth paragraph: perhaps change the two motifs to be F-G-X and X-G-P to make it easier to relate.*

This is a good suggestion and we have changed the text accordingly.

*9) Subsection “Exploring evolutionary contacts for different TatA paralogs”, fourth paragraph: seems odd that E8 isn't discussed here.*

The identity of the position 8 residue is now discussed.

*10) Subsection “Evolutionary co-evolution analysis identifies additional inter-subunit contact sites within the TatBC complex”, last paragraph: can't really see the flexible regions of the periplasmic cap loops.*

To address this issue we have:

Altered Figure 8 to include, in place of the tetramer model, an additional view of the trimer model from the periplasmic side of the membrane showing the periplasmic loops and associated co-evolutionary contacts;

Provided a new supplementary figure (Figure 8—figure supplement 3) containing views of the tetramer model equivalent to those now shown for the trimer model in Figure 8;

Provided as a new supplementary video (Video 3) a rotating model of the trimer to show the periplasmic cap regions and other features of the model that are more easily appreciated in such an animation than in the static images;

Altered Figure 9, so that a trimeric model is shown consistent with the revision to Figure 8.

11) Subsection “Evolutionary co-evolution analysis identifies additional inter-subunit contact sites within the TatBC complex”, last paragraph: the rationale for filling with lipids or water should be discussed. Was anything learned here? It seems there should be some substantial differences. (I realize there are space constraints).

The rationale for these modelling experiments is now discussed together with their potential biological significance.

*12) Figure 2: in the logo plot the length is constrained. How were differences in length accounted for in this presentation.*

We now clarify in the Figure 2 legend that the logo plots cover the same region of the TatA family proteins that is analyzed in Figure 2 and uses the sequence numbering of the *E. coli* TatA and TatB proteins. The *E. coli* TatA and TatB proteins, from residue 1 to 40, were used as reference sequences in the multiple sequence alignment, with (rare) non-aligned residues from the other protein sequences ignored.